# Ni/Mn-Complex-Tethered Tetranuclear Polyoxovanadates: Crystal Structure and Inhibitory Activity on Human Hepatocellular Carcinoma (HepG-2)

**DOI:** 10.3390/molecules28196843

**Published:** 2023-09-28

**Authors:** Fumei Shi, Yilan Chen, Chuanheng Dong, Jiajia Wang, Chunman Song, Yalin Zhang, Zhen Li, Xianqiang Huang

**Affiliations:** 1School of Life Science, Liaocheng University, Liaocheng 252000, China; shifumei@lcu.edu.cn; 2School of Chemistry and Chemical Engineering, Liaocheng University, Liaocheng 252000, China; chenyilanaabb@163.com (Y.C.); chuanhdong@163.com (C.D.); wjjyes2022@163.com (J.W.); s612327980@163.com (C.S.); hxqqxh2008@163.com (X.H.)

**Keywords:** polyoxovanadates, imidazole ligands, human hepatocellular carcinomas, anticancer, apoptosis

## Abstract

Polyoxometalates (POMs) exhibit unique structural characteristics and excellent physical and chemical properties, which have attracted significant attention from scholars in the fields of anticancer research and chemotherapy. Herein, we successfully synthesized and structurally characterized two novel polyoxovanadates (POVs), denoted as POVs-**1** and POVs-**2**, where [M(1-vIM)_4_]_2_[V^V^_4_O_12_]·H_2_O (M: Ni^II^ and Mn^II^, 1-vinylimidazole abbreviated as 1-vIM) serve as ligands. The two POVs are isomeric and consist of fundamental structural units, each comprising one [V_4_O_12_]^4−^ cluster, two [M(1-vIM)_4_]^2+^ cations, and one water molecule. Subsequently, we evaluated the cell viability of human hepatocellular carcinoma (HepG-2) cells treated with the synthesized POVs using the MTT (3-[4,5-dimethylthiazol-2-yl]-2,5-diphenyl tetrazoliumbromide) assay. And the changes in cell nucleus morphology, mitochondrial membrane potential (Δψ_m_), and reactive oxygen species levels in HepG-2 exposed to POVs were monitored using specific fluorescent staining techniques. Both hybrid POVs showed potent inhibitory activities, induing apoptosis in HepG-2 cells along with significant mitochondria dysfunction and a burst of reactive oxygen species. Notably, the inhibitory effects of POVs-**2** were more pronounced than those of POVs-**1**, which is primarily attributed to the different transition metal ions present. These findings underscore the intricate relationship between the metal components, structural characteristics, and the observed antitumor activities in HepG-2 cells.

## 1. Introduction

Cancer is one of the major diseases leading to human mortality, and it poses a grave threat to human well-being. As indicated by the GLOBOCAN 2020 assessments of cancer incidence and mortality presented by the International Agency for Research on Cancer, the year 2020 witnessed a global estimate of 19.3 million new cancer cases and an alarming toll of nearly 10.0 million cancer-related deaths. Projections suggest that the worldwide burden of cancer will soar to 28.4 million cases by 2040, marking a staggering 47% surge from the figures in 2020 [1]. To date, chemotherapy remains one of the most prevalent treatment options for cancer. Nonetheless, chemotherapy consistently gives rise to severe bodily side effects, including weakened immunity and multi-organ failure [2]. Therefore, the quest for innovative anticancer drugs with diminished systemic toxicity persists as a great challenge.

Polyoxometalates (POMs) are a class of nano-sized metal-oxide polyoxoanion clusters, composed of corner- and edge-sharing [MO_x_] polyhedrals, where M generally represents Mo, W, V, Nb, or Ta. POMs, as an adaptable nanoscale multinuclear metal-oxygen cluster, have captured tremendous attention across diverse fields, including biology [3], catalysis [4], energy [5], materials [6,7], and magnetism, due to their well-defined structure and adjustable physicochemical properties [8,9,10]. Among the subclasses of POMs, POVs are of particular interest and have been extensively researched, not only for their significance in homogeneous and heterogeneous catalysis, but also for their potential applications in medicine and bioinorganic chemistry [11,12]. Recent advancements have underscored the potential of POVs against cancer, bacterial resistance, and antiviral infection, a trait attributed to their varied coordination patterns and oxidation states [13,14,15,16,17,18]. For example, Wu’s group engineered functionalized phosphovanadomolybdate, showing notable anticancer activity against human hepatocellular carcinomas (HepG-2) and human breast cancer (MDA-MB-231) at pH = 7.02 [19]. Decavanadate-based complexes have also demonstrated antitumor effectiveness against various human cancer cells, with no adverse impact on normal human cells even at high concentrations. The cytotoxicity value of the V_10_-based complex against A549 was lower compared to that of cisplatin [20]. Nonetheless, due to the structural attributes of POVs, the development of tetranuclear POV framework structures for enhanced anticancer activity has rarely been reported.

In this work, we synthesized two novel inorganic–organic hybrid POV structures, [M^II^(1-vIM)_4_]_2_[V^V^_4_O_12_]·H_2_O (designated as POVs-**1** and POVs-**2**), by utilizing 1-vinylimidazole (1-vIM) as the N-donor ligand to assemble with ammonium metavanadate and various metal ions under standard conditions (Figure 1). The structures were comprehensively characterized using single-crystal X-ray diffraction (SXRD), powder X-ray diffraction (PXRD), Fourier-transform infrared spectroscopy (FT-IR), and elemental analysis (EA). Furthermore, we assessed the antitumor activity of the hybrid POV complexes through DAPI staining, evaluation of their mitochondrial membrane potential (Δψm), and measurement of reactive oxygen species. Remarkably, POVs-**2** demonstrated significantly enhanced antitumor activity.

## 2. Results and Discussion

### 2.1. Structure Analysis

The conventional crystal growth method was used, and the reaction solution was allowed to slowly evaporate at room temperature over the course of one week to yield block crystals of POVs. Specific synthesis steps are described in the synthesis section. Based on the single-crystal X-ray diffraction analysis, the structural formula of POVs-**1** was [Ni^II^(1-vIM)_4_]_2_[V^V^_4_O_12_]·H_2_O, which crystallizes in the *P*2_1_/n space group. All Ni atoms within POVs-**1** exhibited six-coordination in an octahedral geometry, coordinated by four ligands via imidazole N atoms, and the terminal O atom derived from the [VO_4_] tetrahedron. In the [V_4_O_12_]^4−^ anion cluster, all V atoms presented a distorted [VO_4_] tetrahedral geometry and were coordinated by four O atoms (Figure 1a). Two imidazole complexes were arranged alternately and linked by the [V_4_O_12_] cluster to form POVs-**1** with a V···V distance of 5.041 Å. The details of the bond lengths and angles of Ni–N/O, V–O, O/N–Ni–O/N and O–V–O are listed in Appendix A. Of particular interest are the weak hydrogen bonding between ligand molecules, water molecules, and polyanions give rise to two-dimensional (2D) layer structures (Figure 1b). Additionally, the interaction between ligand molecules and adjacent [V_4_O_12_]^4−^ pseudo planes contributed to the formation of a 3D supramolecular structure (Figure 1c). Notably, the vIM ligands of Ni (II) units were retained within the quadrilateral channels by π-π (π-stacking) interactions. Within the 3D networks, the square pores of this framework were found to measure approximately 2.60 Å across, as determined from the largest pink sphere that could fit into the pores, as shown in Figure 1d.

Single-crystal X-ray structural analyses revealed that the cluster units of POVs-**2** and POVs-**1** were isostructural, differing only in the transition metal atom. The structural formula of POVs-**2** is [Mn^II^(1-vIM)_4_]_2_[V^V^_4_O_12_]·H_2_O, crystallizing in the *P*2_1_/n space group. POVs-**2** comprised two Mn^2+^ cations, one [V_4_O_12_]^4−^ anion cluster, and eight 1-vIM ligands. The vanadium atoms within the [V_4_O_12_]^4−^ anion cluster exhibited a distorted [VO_4_] tetrahedral geometry. All manganese atoms in POVs-**2** adopted a six-coordinate octahedral geometry, with coordination involving four ligands via imidazole N atoms around the central Mn atom, the terminal O atom derived from the [VO_4_] tetrahedron. The bond lengths and angles of Mn–N/O, V–O, O/N–Mn–O/N and O–V–O are listed in Appendix A. Similarly, an additional direction of interaction between ligand molecules and adjacent [V_4_O_12_]^4−^ pseudo planes allowed POVs-**2** to form an intriguing 3D supramolecular structure (Figure 2c). Within these 3D networks, four adjacent [VO_4_] tetrahedra were linked through corner-sharing O atoms, creating [V_4_O_12_]^4−^ clusters. Additionally, one terminal O atom from the [VO_4_] tetrahedron engaged in weak interaction to connect with another imidazole ligand subunit, thus contributing to the establishment of a complete 3D supramolecular structure. Notably, compared to POVs-**1**, POVs-**2** exhibited fair-sized cavities. Unlike the V-O-M coordination bond reported in the literature for the 3D structure, the supramolecular structure of POVs-**2** relied solely on weak interactions, devoid of V-O-M coordination bonds. The square pores of this framework measured approximately 2.63 Å across, based on the dimensions of the largest pink sphere capable of fitting into them, as shown in Figure 2d. Crystallographic data and structure refinement for POVs-**1** and POVs-**2** are summarized in Appendix A. The crystallographic data were deposited at the Cambridge Crystallographic Data Center (CCDC) under deposition numbers 2,279,426 (POVs-**1**) and 2,279,427 (POVs-**2**).

As shown in Appendix A, the FT-IR spectra provide valuable information for investigating the structures of POVs-**1** and POVs-**2**. The FT-IR spectra were recorded as KBr pellets in the range of 4000–400 cm^−1^ on a Nicolet 170 SXFT/IR spectrometer. Particularly, in the 1000–500 cm^−1^ region, characteristic bands for M-O_t_ (terminal oxygen atoms) and M-O_b_ (bridging oxygen atoms) can be observed. In particular, the V=O_t_ stretching vibrations appear at 930 cm^−1^ for POVs-**1**. The absorption bands at 797, 655, and 594 cm^−1^ in POVs-**1** can be attributed to the bridging V-O-V or M-O-V stretching vibrations, consistent with the reported structures [21,22]. The pronounced absorption bands in the range of 1600 cm^−1^ to 1450 cm^−1^ are assigned to C=C and C=N vibrations (1-vIM ligand). Moreover, the absorption bands of coordinated and free water molecules in POVs-**1** appear around 3500–3300 cm^−1^ [23]. The simulated and experimental PXRD patterns of POVs-**1** and POVs-**2** are shown in Appendix A, respectively. The diffraction peak positions of the experimentally obtained PXRD patterns of POVs-**1** and POVs-**2** closely align with those of the simulated single-crystal patterns.

### 2.2. Pharmacology Evaluation

To investigate whether the developed complex would similarly exert an impact on cell viability, cells were exposed to various concentration complex for the indicated duration. The MTT assay was then conducted to reveal the potential effects of the complex on cell viability. The results show that the complex effectively inhibits cell proliferation, as evinced by a noticeable decrease in cell count and diminished cell viability (Figure 3). These findings indicate that the complex exhibited dose- and time-dependent cytotoxicity in the tested HepG-2 cells. The *IC*_50_ values of the complex were determined from three independent experiments in the stressed cells group, resulting in values of 60 ± 5.45 μM POVs-**1** and 40 ± 6.51 μM for POVs-**2**. Compared with data from the literature, especially TBA-V_10_ complexes, POVs-**2** exhibited superior anticancer activity [24]. The distinct antiproliferative capacity of the complexes can undoubtedly be attributed to their unique spatial structure, with POVs-**2** displaying stronger inhibitory abilities. Moreover, characteristic morphological changes in the cells were observed under inverted microscopy.

The morphology of the responsive cells was highly distorted, with the emergence of membrane bubbles, cellular debris, cell folding, and shrinkage in treated cells. Most chemotherapeutic agents exerted potent anticancer activity by inducing cell death through various approaches such as apoptosis, suicide, or necrosis, each marked by distinctive traits. To identify the mode of cell death, nuclei co-stained with DAPI were utilized in the subsequent cytotoxicity assay to differentiate the type of apoptosis in HepG-2 cells responding to the assault using fluorescence-activated cell sorting. Both complexes, POVs-**1** and POVs-**2**, displayed notable bright blue fluorescence indicative of condensed nuclei and nuclear granules (Figure 4). These features arise from fragmentary DNA and are emblematic of apoptosis, in contrast to the even and weak blue fluorescence observed in the control group. Moreover, the POVs-**2** treated groups exhibited a significant rise in the population of apoptotic cells compared to the control and POVs-**1** groups. Furthermore, the intensified bright blue fluorescence and the percentage of apoptotic cells in the POVs-**2** treated groups were notably diminished in the presence of NAC (N-Acetylcysteine), a classic antioxidant. This implies that both complexes, POVs-**1** and POVs-**2,** induce apoptosis in HepG-2 cells by triggering intrinsic reactive oxygen species (ROS), with POVs-**2** demonstrating stronger effects than POVs-**1**. The initiation of apoptosis in cancer cells is considered highly valuable both in therapy and cancer prevention with this exploration. 

Mitochondria play a unique role in numerous crucial physiological cellular procedures, including signal transduction, proliferation, differentiation, cell cycle regulation, cell death, calcium homeostasis, and cellular redox status regulation [25]. Apoptosis initiation involves several mechanisms and pathways, with the mitochondria’s dysfunction-mediated intrinsic apoptotic pathway being pivotal in the cell-death process [26]. Mitochondria have been identified as valuable targets for cancer therapy owing to their primary function in energy supply and cellular signal regulation [27], which play multiple crucial roles in intrinsic programmed cell death [28]. The integrity of mitochondrial is essential for maintaining their physiological functions as cellular energy suppliers. Disruption of this integrity through processes such as membrane degradation and depolarization of mitochondria outer membrane can lead to a breakdown in the energy chain [29]. Depolarization of the mitochondrial membrane is a critical event in response to external stimuli, leading to the loss of mitochondrial membrane potential (ΔΨ_m_), which subsequently promotes excessive permeability of the outer membrane and uncouples the oxidative phosphorylation respiration chain, halting energy production [30]. To investigate whether the complex triggers mitochondrial devastation to induce cell suicide, we examined the mitochondrial dysfunction and membrane alterations in response to the complexes. As shown in Figure 5a, the mitochondrial membrane potential (ΔΨ_m_) was assessed using the specific fluorescence switch of the JC-1 probe. In healthy cells, JC-1 accumulates in the mitochondria cells as aggregates emitting red fluoresce. However, in treated cells, JC-1 remains in the cytoplasm in its monomeric form, emitting green fluoresce, indicating the collapse of the mitochondrial membrane potential (ΔΨ_m_). This transition from red to green fluorescence reflects JC-1’s shift from an aggregator at normal ΔΨ_m_ to a monomer at collapsed ΔΨ_m_. Depolarized or dysfunctional mitochondria, with reduced membrane potential, fail to uptake JC-1 and form aggregates. Similarly, the disturbed red and green fluorescence of mitochondria induced by complex POVs was significantly alleviated in the presence of NAC, which indicated that both complexes, POVs-**1** and POVs-**2**, triggered this loss of ΔΨ_m_, a crucial aspect of this investigation (Figure 5b). This loss leads to the release of various apoptogenic factors from the mitochondria into the cytoplasm and disturbs mitochondrial energy metabolism, inducing an ROS burst through proton decoupling by free radicals in the respiratory chain [31,32,33,34,35].

Increasing ROS levels in cells represents an optional therapeutic strategy for cancer [36]. Oxidative stress provokes the generation of and increase in cellular ROS, ultimately initating a cascade of pathogenic events [37]. Mitochondria stand as the main sources of ROS in cells [38]. Research has indicated a strong correlation between mitochondrial reactive oxygen species (mROS) and the occurrence and adverse outcomes of various diseases [39].

Impairment of the mitochondria through small chemical molecules can lead to irreversible ROS generation [40], which, in turn, can further compromise mitochondrial integrity through an ensuing ROS burst [41]. It is well-documented that several anti-cancer drugs induce ROS generation, causing DNA damage, activation of apoptosis, ATP down-regulation, decreased mitochondrial membrane potential, release of Cyt-c into the cytoplasm, and activation of Cas3/9 [42,43,44]. Considerable efforts have focused on exploiting mitochondria-targeting ROS generators through the use of targeting agents [45]. To determine whether the complex regulates intracellular ROS levels in HepG-2 cells, we treated the cells with POVs (24 h, 60 μM) and then exposed them to H_2_DCFDA dye, followed by measuring ROS levels using inverted microscopy. The abundance of green fluorescent intensity in the ROS levels was anticipated, as is consistent with the induction of POVs. This is in contrast to the intact cell group (Figure 5), which exhibited scarce green fluorescent signals, indicating minimal ROS generation. Thus, this study provides evidence that the invasive POVs lead to ROS generation and elevated oxidative stress in disturbed cells. To further elucidate the role of ROS in this context, NAC, a typical ROS scavenger, was applied to the treated sample. The collected data showed an inconspicuous green fluorescence intensity, indicating a remarkable reduction in ROS in the tested cells. NAC effectively mitigated the vulnerability of fragile cells by suppressing ROS initiation provoked by POVs (Figure 6a,b), and a similar trend was observed in MMP (Figure 5a). The reduced susceptibility to drug-mediated ROS levels and attenuated mortality further confirmed that oxidative stress was responsible for the cytotoxicity of the complex. The results validated that a mitochondrial-targeted ROS-associated profile was responsible for the lethal consequence resulting from the complex stimulation in HepG-2 cells. For approval as a drug, the development of potential complexes for therapeutic treatment requires comprehensive investigation. These studies demonstrate the potential applications of such inorganic–organic hybrid POVs, making them promising options for therapeutic interventions. 

## 3. Materials and Methods

### 3.1. General Methods

All reagents and solvents for synthesis were purchased from commercial sources and used without further purification. The reagents and chemicals, including vanadium pentoxide (99.0%), nickel chloride hexahydrate (98.0%), manganese chloride tetrahydrate (99.0%), 1-vinylimidazole (99.0%), and tetramethylammonium hydroxide solution (AR, 25% in water), were all purchased from Aladdin Chemical. FT-IR spectra were recorded in the range of 4000–400 cm^−1^ on a Nicolet 170 SXFT/IR spectrometer. The C, H, and N elemental analyses were conducted on a Perkin-Elmer 240C elemental analyzer. The crystallographic data can be obtained free of charge from The Cambridge Crystallographic Data Center via www.ccdc.cam.ac.uk/data_request/cif (accessed on 5 July 2023).

### 3.2. Synthesis of POVs-1 and POVs-2

Preparations of [Ni^II^(1-vIM)_4_]_2_[V^V^_4_O_12_]·H_2_O (POVs-**1**): V_2_O_5_ (0.27 mmol, 0.049 g) and NiCl_2_·6H_2_O (0.25 mmol, 0.059 g) were added into 10 mL of water. The mixture was then stirred, along with a 25% aqueous solution of tetramethylammonium hydroxide (1.09 mmol, 0.1 g) and 1-vinylimidazole (22.08 mmol, 2.08 g), for 1 h. Subsequently, the mixed solution was heated and stirred at 90 °C for 72 h. The resulting solution was then subjected to thermal filtration, and the obtained filtrate was allowed to slowly evaporate at room temperature over the course of one week, resulting in the formation of pale blue block crystals of POVs-**1**. Crystals suitable for SXRD analysis were obtained after leaving the solution undisturbed for two weeks. The overall yield was 53% (based on V). The calculated (found) elemental composition for C_40_H_50_Ni_2_N_16_O_15_V_4_ was as follows: C, 36.50 (36.39); H, 3.80 (3.93); N, 17.03 (16.87).

Preparation of [Mn^II^(1-vIM)_4_]_2_[V^V^_4_O_12_]·H_2_O (POVs-**2**): The synthetic procedure for POVs-**2** was similar to that of POVs-**1,** with the exception that NiCl_2_·6H_2_O (0.25 mmol, 0.059 g) was used instead of MnCl_2_·4H_2_O (0.25 mmol, 0.049 g). The resulting brown crystals, which were suitable for SXRD analysis, were obtained after allowing the solution to stand undisturbed for two weeks. The overall yield was 55% (based on V). The calculated (found) elemental composition for C_40_H_50_Mn_2_N_16_O_15_V_4_ was as follows: C, 36.71 (36.58); H, 3.82 (3.97); N, 17.13 (17.01).

### 3.3. Cell Culture

HepG-2 cells were sourced from the American Type Culture Collection and maintained in Dulbecco’s modified Eagle’s medium (DMEM, Gibco, Invitrogen, Grand Island, NE, USA), supplemented with 10% fetal bovine serum (FBS, Gibco), 100 U/mL penicillin, and 100 µg/mL streptomycin. The cells were incubated at 37 °C in a humidified atmosphere containing 5% CO_2_, and the medium was refreshed every 2 days. Upon seeding, the cells were allowed to incubate overnight to achieve the desired confluence. Subsequently, they were treated with different concentrations of complexes, POVs-**1** and POVs-**2**, for 24 h after pretreatment with or without NAC for 0.5 h. 

### 3.4. Determination of Cell Viability

Both synthesized complexes, POVs-**1** and POVs-**2**, were dissolved in DMSO to create a 10 mM stock solution for subsequent dilution into working concentrations. 

HepG-2 cell viability was assessed using the 3-(4,5-dimethylthiazol-2-yl)-2,5-diphenyltetrazolium bromide (MTT) assay. Cells were seeded in a 96-well plate at a density of 1 × 10^5^ cells mL^−1^ and cultured with varying concentrations of the complex, a combination of NAC/complexes, or in the absence of treatment in a dose- and time-dependent manner. The concentration ranged from 10 μM to 100 μM, and the incubation times were 24 h and 48 h. The culture medium used was DMEM medium (Hyclone) is supplemented with 10% FCS, 2 mM L-glutamine, 100 U mL^−1^ penicillin, and 100 µg mL^−1^ streptomycin. The cells were cultured in a humidified atmosphere with 5% CO_2_ at 37 °C, and a native group was included as a negative control.

To perform the MTT assay, a solution of MTT (5 mg mL^−1^) was added to each well and incubated for an additional 4 h at 37 °C with 5% CO_2_. After incubation, the supernatant was carefully discarded, and 100 μL DMSO was added to each well and mixed thoroughly to dissolve the formazan crystals. The absorbance of the plate was measured at 490 nm using a multi-well plate reader (Agilent synergy H1). The percentage of cell viability was calculated using the following formula: % Viability = [OD of treated cells−OD of blank/OD of control cells−OD of blank] × 100. All experiments were independently conducted in triplicate.

### 3.5. DAPI Staining of Cells

A total of 1 × 10^5^ cells were cultured in a six-well plate and treated with either POVs-**1** or POVs-**2** for 24 h. Subsequently, the cells were fixed with 4% methanol for 30 min at room temperature and then rinsed twice with PBS buffer. Following this, the cells were stained with a 100 μL solution of DAPI (5 µg/mL) for 10 min in the dark. Then cells were then washed with PBS and examined under a Nikon Ti inverted fluorescent microscope equipped with NIS-Elements software D for further analysis of the images. Cells displaying condensed and fragmented nuclei, with enhanced fluorescence upon DAPI staining, were identified as apoptotic cells [46]. The experiments were repeated three times. 

### 3.6. Assessment of Mitochondrial Membrane Potential (Δψ_m_)

A total of 1 × 10^5^ cells were treated with either POVs-**1** or POVs-**2** for 24 h. Following the treatment, the cells were washed twice with PBS and then suspended in a mitochondrial incubation buffer. JC-1, a lipophilic molecular probe that specifically penetrates the cell and accumulates in the mitochondria, was added to the cells at a final concentration of 10 nM. The cells were then incubated at 37 °C in the dark for 30 min. Afterwards, the mitochondrial membrane potential was determined using an inverted fluorescence microscope (Nikon Ti Japan) at an excitation wavelength of 495 nm and emission maxima at 514–529 nm for the monomer form, and at 585–590 nm for the J-aggregate form, as per the protocol. The intensity of the fluorescence was analyzed using Image J, and further image analysis was performed using the coupled NIS-Elements software of the Nikon Ti inverted fluorescent microscope. These experiments were repeated three times.

### 3.7. Measurement of Intracellular Reactive Oxygen Species (ROS)

To assess the ability of complexes POVs-**1** or POVs-**2** to trigger ROS in HepG-2 cells, the cell-permeable fluorescent probe H_2_DCFDA was utilized. In brief, a total of 1 × 10^5^ cells were treated with complex POVs-**1** or POVs-**2,** or a combination of NAC and the complex in a CO_2_ incubator for 24 h. Subsequently, the medium was replaced with fresh medium containing 5 μg/mL of H_2_DCFDA, and the cells were further incubated for 30 min. After incubation, the cells were rinsed twice with chilled PBS. The green fluorescence intensity within the cells was then examined using an inverted fluorescence microscope (Nikon Ti, Tokyo, Japan) equipped with excitation and emission filters set at 492–495 and 517–527 nm, respectively, as per the protocol. The fluorescence intensity was quantified with Image J. These experiments were repeated three times. 

### 3.8. Statistical Analyses

The results are expressed as the mean ± standard errors (SEM). All analyses were conducted using specialized software. In order to more intuitively and clearly present the percentage of apoptosis cells induced by complexes, apoptosis cells were counted and analyzed via image processing software Image J 1.53t and supporting software in three parallel random samplings of the experimental results graph. Significant differences among the treatment effects were determined using one-way ANOVA, followed by Tukey’s post hoc test for multiple comparisons, with statistical significance set at *p* < 0.05 via IBM SPSS Statistics 27 software.

## 4. Conclusions

In this study, we report two intriguing POVs with distinct metal coordination modes, specifically the classical six-coordination mode. Notably, this metal coordination pattern exhibits a significant inhibitory effect on HepG-2 cells. Both [Ni^II^(1-vIM)_4_]_2_[V^V^_4_O_12_]·H_2_O (POVs-**1**) and [Mn^II^(1-vIM)_4_]_2_[V^V^_4_O_12_]·H_2_O (POVs-**2**) demonstrated robust antitumor effects against HepG-2 cells, achieved through the induction of a mitochondrial-targeted intracellular ROS burst. This intriguing finding suggests their promising potential as candidates for the development of anti-cancer drugs. Moreover, the hybrid POVs-**2** showed even greater efficacy compared to POVs-**1**, indicating a pivotal role played by the introduction of the Mn element, which confers superior biological activity compared to the Ni element. Therefore, further investigations, such as an analysis of the candidate caspase to elucidate the underlying mechanism, as well as additional experiments to accurately determine the *IC*_50_ value, thus assessing the degree of HepG-2 cell tolerance, are highly warranted. Furthermore, considering the potential clinical applications, it becomes imperative to conduct toxicological studies involving exposure to lower doses of POMs and their related complexes, especially in the context of in vivo CT imaging. This work provides a new perspective on the development of novel anticancer drugs by introducing bioactive Mn elements into the atomic level structure of POVs. 

## Data Availability

Data will be available on request.

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
