# Peer review of "Ni/Mn-Complex-Tethered Tetranuclear Polyoxovanadates: Crystal Structure and Inhibitory Activity on Human Hepatocellular Carcinoma (HepG-2)"

_molecules, 2023, doi:10.3390/molecules28196843_

Round 1

Reviewer 1 Report (New Reviewer)

The article may be published in Molecules after correcting the following remarks: 1) Please indicate the degree of oxidation of metals in the text of the article. 2) Supplementary: According to the PXRD of the POV-1 complex (Fig. S3), the sample
has admixtures. The red (experiment) diagram has additional peaks compared
to the black one. 3) The units of measurement on the IR spectra are not indicated.
The main vibration frequencies in the experimental part should be
given in the main text of the article, in addition, it should be
indicated that the IR spectra were recorded from tablets with KBr. 4) The X-ray diffraction analysis should contain a link to the CCDC site. 5) In the text, there is a selection of individual fragments in yellow,
this should be removed.
6)
In the experimental part, there is no enumeration of reagents - purity,
manufacturers. In particular, tetraethylammonium hydroxide is supplied by
manufacturers as a solution, but the text does not indicate which one the
authors used. It is necessary to indicate not only all masses (including
the yield in grams), but also the amount of substance of the reagents used.
In the experimental part, it is necessary to indicate the full formulas of
compounds 1 and 2, and not just their symbols from the text of the article.
7) Complexes 1 and 2 are coordination polymers, this must be reflected in
the formula and description. The terminal oxygen in coordination environment
of nickel and manganese obviously belongs to the neighboring
polyoxovanadate fragment of polymer chain, this should be reflected
in the text. In the description of structures 1 and 2, the authors
indicated “surface O atom”, while based on the formula of the complex,
there is no such atom. 8) How were spheres modeled in the pores of the complexes? I recommend
specifying the volume free cell space.
9) Figures 1, 2, scheme 1: not only hydrogen atoms, but also solvate
molecules are not shown. This should be indicated in the figure captions. 10) In the conclusions, more detailed information about the compounds
(formulas, a very brief description) should be given and it should be
emphasized with which metal the higher activity was found.
11) Complexes of very similar structure were obtained earlier, the authors
should provide references to them. Authors should compare the anticancer
activity of complexes 1 and 2 not only with polyoxovanadates, but also
with vinylimidazole, and known anticancer drugs.

Author Response

September 21th, 2023

Subject: Resubmission of a manuscript for molecules-Manuscript ID: molecules- 2604251

Title: Ni/Mn-complex-tethered tetranuclear Polyoxovanadates: crystal structure and inhibitory activity on human hepatocellular carcinoma (HepG-2)

Author: Fumei Shi1, Yilan Chen2, Chuanheng Dong2, Jiajia Wang2, Chunman Song2, Yalin Zhang2,*, Zhen Li2,* and Xianqiang Huang2

Dear Molecules Editorial Office:

We thank you and the referees for your comments on this manuscript entitled "Ni/Mn-complex-tethered tetranuclear polyoxovanadates: crystal structure and inhibitory activity on human hepatocellular carcinoma (HepG-2)" (molecules-2604251). We are happy that they find the content of quality and novelty to ensure its suitability as a full paper in Molecules. Please find enclosed a revision of the manuscript, referenced above. We appreciate the reviewer’s constructive comments and suggestions to improve the manuscript. In response, we have done our best to resolve all the issues as well as revising the language to enhance the quality of this work.

In the revised version, we have addressed the points that the referees made. Point-by-point responses in the order the comments appeared in your letter and in the reviewer’s comments are provided below with the corresponding edits that we made to the manuscript and supporting information. These edits are also highlighted in yellow in both the manuscript and supporting information to reflect the changes.

Moreover, the manuscript has undergone editing for proper English language, grammar, punctuation, spelling and overall style by the highly qualified English speaking editor at EditRun.

We wanted to thank you and the referees for the insightful comments and for taking the time to provide critical input. We believe this manuscript is improved by this process, and hope you agree.

Best regards,

Yours Sincerely,

Zhen Li

lizhenlcu@163.com

School of Chemistry and Chemical Engineering,

Liaocheng University

The list of changes or comments is shown as following:

#Reviewer 1

Comments:

The article may be published in Molecules after correcting the following remarks: Response: We are very pleased and excited by positive comments on the novelty and significance of our study. We also thank the reviewer’s detailed technical comments/suggestions, which we have addressed point-by-point below.

  1. Please indicate the degree of oxidation of metals in the text of the article.

Response: Thank you for insightful suggestion. We have added the degree of oxidation of metals in the text of the article. The valence state of vanadium is +5, and the valence state of Mn/Ni is +2 in the complexes, respectively.

  1. Supplementary: According to the PXRD of the POV-1 complex (Fig. S3), the sample has admixtures. The red (experiment) diagram has additional peaks compared to the black one.

Response: Thank you for insightful suggestion. We have retested the PXRD experiment of the POVs-1 complex, and the PXRD patterns have added in the Figure S3. According to the PXRD of the POVs-1, the diffraction peak positions of the experimentally obtained PXRD patterns of POVs-1 are consistent with the simulated diffraction peaks. The reason why some peaks are not obvious may be that the crystal grinding is not sufficient during the test, or the test signal is weak.

Figure S3. The PXRD patterns of POVs-1 (Red) and calculated data (Black).

  1. The units of measurement on the IR spectra are not indicated. The main vibration frequencies in the experimental part should be given in the main text of the article, in addition, it should be indicated that the IR spectra were recorded from tablets with KBr.

Response: Thank you for insightful question. The units of measurement on the IR spectra have added Figure S1 and S2 in the supplementary materials. We have added the main vibration frequencies in the results and discussion part and the IR spectra experimental part have added in the article. The FT-IR spectra were recorded as KBr pellets in the range of 4000-400 cm−1 on a Nicolet 170 SXFT/IR spectrometer. Particularly, in the 1000-500 cm−1 region, characteristic bands for M-Ot (terminal oxygen atoms) and M-Ob (bridging oxygen atoms) are observed. Especially, the V=Ot stretching vibrations appear at 930 cm-1 for POVs-1. The absorption bands at 797, 655 and 594 cm-1 in POVs-1 can be attributed to the bridging V-O-V or M-O-V stretching vibrations, consistent with reported structures. The pronounced absorption bands in the range of 1600 cm-1 to 1450 cm-1 are assigned to C=C and C=N vibrations (1-vIM ligand). Moreover, the absorption bands of coordinated and free water molecules in POVs-1 appear around 3500-3300 cm-1. Please see lines 128-138, page 4.

Figure S1. The FT-IR spectrum of POVs-1.

Figure S2. The FT-IR spectrum of POVs-2.

  1. The X-ray diffraction analysis should contain a link to the CCDC site.

Response: Thank you for insightful suggestion. We have added the CCDC number of the POVs-1 and POVs-2 in the X-ray diffraction analysis part. Crystallographic data and structure refinement for POVs-1 and POVs-2 are summarized in Table S1. The crystallographic data in this article have been deposited at the Cambridge Crystallographic Data Center (CCDC) under deposition numbers CCDC 2279426 (POVs-1) and 2279427 (POVs-2). Please see lines 122-124, page 4. Both crystallographic data can be obtained free of charge from The Cambridge Crystallographic Data Center via www.ccdc.cam.ac.uk/data_request/cif.

  1. In the text, there is a selection of individual fragments in yellow, this should be removed.

Response: Thank you for insightful suggestion. We have removed the individual fragments in yellow.

  1. In the experimental part, there is no enumeration of reagents - purity, manufacturers. In particular, tetraethylammonium hydroxide is supplied by manufacturers as a solution, but the text does not indicate which one the authors used. It is necessary to indicate not only all masses (including the yield in grams), but also the amount of substance of the reagents used. In the experimental part, it is necessary to indicate the full formulas of compounds 1 and 2, and not just their symbols from the text of the article.

Response: Thank you for insightful suggestion. We have added the purity and manufactures about of reagents in the experimental part. The reagents and chemicals, including vanadium pentoxide (99.0%), nickel chloride hexahydrate (98.0%), manganese chloride tetrahydrate (99.0%), 1-vinylimidazole (99.0%) and tetramethylammonium hydroxide solution (AR, 25% in water), were all purchased from Aladdin Chemical. We have corrected the experimental part, please see the 3.2 Synthesis of POVs-1 and POVs -2.

Preparation of [NiII(1-vIM)4]2[VV4O12]·H2O (POVs-1): V2O5 (0.27 mmol, 0.049 g) and NiCl2·6H2O (0.25 mmol, 0.059 g) were added to 10 mL of water. The mixture was then stirred along with a 25% aqueous solution of tetramethylammonium hydroxide (1.09 mmol, 0.1g) and 1-vinylimidazole (22.08mmol, 2.08g) for 1 h.

Preparation of [MnII(1-vIM)4]2[VV4O12]·H2O (POVs-2): The synthetic procedure for POVs-2 was similar to that of POVs-1 with the exception that NiCl2·6H2O (0.25 mmol, 0.059 g) was used instead of MnCl2·4H2O (0.25 mmol, 0.049 g).

  1. Complexes 1 and 2 are coordination polymers, this must be reflected in the formula and description. The terminal oxygen in coordination environment of nickel and manganese obviously belongs to the neighboring polyoxovanadate fragment of polymer chain, this should be reflected in the text. In the description of structures 1 and 2, the authors indicated “surface O atom”, while based on the formula of the complex, there is no such atom.

Response: Thank you for insightful suggestion. POVs-1 and POVs-2 are single complexes, the supramolecular structures are relied solely on weak interactions without covalent bonds. We have added about the terminal oxygen in coordination environment of nickel and manganese belongs to the neighboring polyoxovanadates fragment to the part of Structure Analysis. We have deleted the words about "surface O atom".

  1. How were spheres modeled in the pores of the complexes? I recommend specifying the volume free cell space.

Response: Thank you for insightful suggestion. The spheres in the three-dimensional structure diagram of POVs-1 and POVs-2 are drawn by inserting a pseudoatom into four adjacent transition metal atoms (Mn or Ni) as vertices. Within the 3D networks, the square pores of the framework were found to measure approximately 2.60 Å and 2.63 Å across, respectively, as shown in Figure 1d and Figure 2d.

  1. Figures 1, 2, scheme 1: not only hydrogen atoms, but also solvate molecules are not shown. This should be indicated in the figure captions.

Response: Thank you for insightful suggestion. We have added the instruction "All hydrogen atoms and solvent molecules are omitted for clarity." to the figures’ captions.

  1. In the conclusions, more detailed information about the compounds (formulas, a very brief description) should be given and it should be emphasized with which metal the higher activity was found.

Response: Thank you for insightful suggestion. We have added the formulas about the compounds to the conclusions. And the Mn metal have higher activity than Ni metal. The sentences "The hybrid POVs-2 has shown even greater efficacy compared to POVs-1, which indicate a pivotal role played by the introduction of the Mn element, which confers superior biological activity compared to the Ni element" are added in the conclusions.

  1. Complexes of very similar structure were obtained earlier, the authors should provide references to them. Authors should compare the anticancer activity of complexes 1 and 2 not only with polyoxovanadates, but also with vinylimidazole, and known anticancer drugs.

Response: Thank you for insightful suggestion. We have provided the complexes of very similar structure were obtained earlier to the article, please see Ref 19 in the part of structure analysis. According to the reported literature, the imidazole compounds have certain anticancer activity against HepG2 or other cancer cells (Eur. J. Med. Chem., 2022, 232, 114188; Molecules 2019, 24, 3259). According to our experimental results, POVs-2 has better anticancer activity than imidazole ligands.

Reviewer 2 Report (New Reviewer)

The complexes are a polymer of polyoxovanadate anions and transition metal coordination cations. It is not dissolve in the state of a polymer. When the complexes are dissolved in DMSO, it is dissociated and separated into two species, that is the cations, M(1-vlM)4X2 and anions, A4V4O12. To evaluate the activity, it is necessary to study the control using the anion only, and the cation complex only, with a combination of a counter ion. Author should evaluate the results with the comparison with the control experiment. The control in the current manuscript is not explained well.

For the coordination modes of V4O12 anions, there are a good reference in Coord. Chem. Rev. 255, 2270-2280 (2011).

The compounds are polymers of V4O12 anion and it should be highly insoluble because of it.  After dissolution in DMSO, the anions and cations may be dissociated and the biological activity may be the combined effects from the anion and cations.

I see no big issue in this paper

Author Response

#Reviewer 2

Comments:

The complexes are a polymer of polyoxovanadate anions and transition metal coordination cations. It is not dissolve in the state of a polymer. When the complexes are dissolved in DMSO, it is dissociated and separated into two species, that is the cations, M(1-vlM)4X2 and anions, A4V4O12. To evaluate the activity, it is necessary to study the control using the anion only, and the cation complex only, with a combination of a counter ion. Author should evaluate the results with the comparison with the control experiment. The control in the current manuscript is not explained well.

Response: Thanks for your insightful suggestion. POVs-1 and POVs-2 are single complexes, the supramolecular structures are relied solely on weak interactions without covalent bonds. According to the reported literature, both cation complex and anion cluster have certain anticancer activity against HepG-2 (J. Inorg. Biochem., 2015, 142: 109-117; J. Mater. Chem., 2005, 15, 4773-4782; Molecules 2018, 23, 1232). The anticancer activity of cationic [Ni(1-vIM)4] and anion [V4O12] on HepG-2 and their specific mechanisms of action will be discussed in subsequent studies.

  1. For the coordination modes of V4O12 anions, there are a good reference in Coord. Chem. Rev. 255, 2270-2280 (2011).

Response: Thank you for insightful suggestion. We have included the reference about the coordination modes of V4O12 anions in the article, please the Ref 12.

  1. The compounds are polymers of V4O12 anion and it should be highly insoluble because of it. After dissolution in DMSO, the anions and cations may be dissociated and the biological activity may be the combined effects from the anion and cations.

Response: Thank you for insightful suggestion. DMSO is very stable and the boiling point is above with 150 °C, which is difficult to decompose under general temperature conditions and does not volatilize. DMSO is almost a universal solvent for cytology research. After dissolution in DMSO, the anions and cations may be dissociated and the biological activity may be the combined effects from the anion and cations. And then, anticancer activity of cationic [Ni(1-vIM)4] and anion [V4O12] on HepG-2 and their specific mechanisms of action will be discussed in subsequent studies.

Reviewer 3 Report (New Reviewer)

The authors describe the synthesis and full characterization of two polyoxovanadate compounds. Also the cell viability of human hepatocellular carcinoma cells was assessed by their treatment with the title compounds. The obtained results have the potential to be applied in the development of anticancer drugs. I find this study well structured and complete. I recommend it for publication in Molecules after a minor revision. I suggest to include in the Introduction the following work, dealing with fully oxidized POVs and their biocompatibility with HeLa cells: Chem. Mater. 2023, 35, 14, 5447–5457 (https://doi.org/10.1021/acs.chemmater.3c00776)

It can be improved.

Author Response

#Reviewer 3

Comments:

The authors describe the synthesis and full characterization of two polyoxovanadate compounds. Also the cell viability of human hepatocellular carcinoma cells was assessed by their treatment with the title compounds. The obtained results have the potential to be applied in the development of anticancer drugs. I find this study well structured and complete. I recommend it for publication in Molecules after a minor revision. I suggest to include in the Introduction the following work, dealing with fully oxidized POVs and their biocompatibility with HeLa cells: Chem. Mater. 2023, 35, 14, 5447–5457 (https://doi.org/10.1021/acs.chemmater.3c00776)

Response: Thank you for insightful suggestion. We have included the reference about dealing with fully oxidized POVs and their biocompatibility with HeLa cells in the article, please the Ref 12.

Reviewer 4 Report (New Reviewer)

The manuscript by Fumei Shi and co-workers describes the synthesis of two new tetravanadates containing 1-vinylimidazole nickel(II) and manganese(II) complexes as ligands, which were characterized by SC-DRX, PXRD, FT-IR and elemental analysis. In addition, anticancer activity in vitro of the human hepatocellular carcinoma cell line (HepG-2) was investigated, showing a higher inhibitory activity for POVs-2 than for POVs-1 and it was correlated with the second metal transition.  In a general sense, this manuscript is interesting because the authors correlate the composition of POVs with changes in cell nuclei morphology, mitochondrial membrane potential (Δψm), and reactive oxygen species levels in cells. The experiments were well-designed, carefully performed and described in detail. The results are clearly presented, and thoroughly discussed, but I missed comparison with other vanadium compounds or other POMs previously reported. The authors only mentioned TBA-V10 in line 151 (reference 22, https://doi.org/10.1016/j.poly.2018.08.052). They should also compare with a commercial drug as 5-fluorouracil.  The authors should be encouraged to run speciation studies in their further studies. 

Author Response

#Reviewer 4

Comments:

The manuscript by Fumei Shi and co-workers describes the synthesis of two new tetravanadates containing 1-vinylimidazole nickel(II) and manganese(II) complexes as ligands, which were characterized by SC-DRX, PXRD, FT-IR and elemental analysis. In addition, anticancer activity in vitro of the human hepatocellular carcinoma cell line (HepG-2) was investigated, showing a higher inhibitory activity for POVs-2 than for POVs-1 and it was correlated with the second metal transition.  In a general sense, this manuscript is interesting because the authors correlate the composition of POVs with changes in cell nuclei morphology, mitochondrial membrane potential (Δψm), and reactive oxygen species levels in cells. The experiments were well-designed, carefully performed and described in detail. The results are clearly presented, and thoroughly discussed, but I missed comparison with other vanadium compounds or other POMs previously reported. The authors only mentioned TBA-V10 in line 151 (reference 22, https://doi.org/10.1016/j.poly.2018.08.052). They should also compare with a commercial drug as 5-fluorouracil. The authors should be encouraged to run speciation studies in their further studies.

Response: Thank you for your comments and insightful suggestion. According to the reported literature, the results of MTT assay showed that 5-fluorouracil as commercial drug were good antitumor against HepG-2 (IC50=9.12 μM). Hence, the anticancer activity of the POVs-1 and POVs-2 showed slightly lower than commercial drug as 5-fluorouracil in terms of IC50 values. In our further study, we will continue to run speciation studies and the results will be reported timely.

This manuscript is a resubmission of an earlier submission. The following is a list of the peer review reports and author responses from that submission.

Round 1

Reviewer 1 Report

The article entitled “Two Imidazole-decorated Tetranuclear Polyoxovanadates Clusters: synthesis, crystal structures and human hepatocellular carcinomas (HepG-2) inhibitory activities” by Fumei Shi with co-authors is devoted to the preparation of two new coordination polymers based on tetranuclear polyoxovanadate (POV) [V4O12]4–, Ni(II) or Mn(II) cations and 1-vinylimidazole (1-vIM) ligands. The authors characterized the compounds obtained using single-crystal structural analysis, infrared spectroscopy and elemental analysis. In addition, some biological properties of the coordination polymers, named as POVs-1 and POVs-2, were evaluated. In general, the article is interesting and can be published in Molecules, but it has a number of serious flaws that need to be corrected before publication.

First of all, tetranuclear polyoxovanadates (referred to as Tetranuclear Polyoxovanadates Clusters in the title) and polyoxometalates (POMs) (page 1, line 40) are not cluster metal cluster complexes. According to F.A. Cotton (Inorg. Chem., 1964, 3, 1217), metal cluster complexes have covalent metal-metal bonds. POMs are polyatomic ions consisting of three or more transition metal oxyanions linked together by shared oxygen atoms. The term cluster is used incorrectly in the article. Also, POVs-1 and 2 are not imidazole-decorated POMs since imidazole-decorated POM is when imidazole coordinates to POM with formation of V-N bonds. Therefore, the title and some parts of the introduction need to be changed.

Second, the main purpose of the article, according to the authors, is to study the biological effects of the obtained compounds, but this looks strange. In the abstract the authors write that “Polyoxovanadates (POVs), as emerging biologically active inorganic metal oxides, have been selected as promising antitumor alternatives in suppression of tumor growth because of low toxicity towards the human body.” Where are the references for such a loud statement? I've never heard that POMs are non-toxic to the human body. Moreover, according to Introduction parts and the results obtained, POVs is quite biologically active and cytotoxic. So, it is not correct to announce that POVs have “low toxicity towards the human body”.

In addition, since the compounds POVs-1 and POVs-2 have been prepared for the first time, it is necessary to describe them in detail and to compare them with similar compounds at least form Ref. 19. There the authors also obtained compound similar compounds of Ni and Cu, namely [Ni2(1-vIM)7H2O][V4O12]·H2O and [Cu2(1-vIM)8][V4O12]·H2O. Moreover, the compound [Cu2(1-vIM)8][V4O12]·H2O has a similar formula to POVs-1 and 2 prepared here, but in a different crystal packing. The authors should compare and discuss the differences in synthesis and crystal structure between them.

The most incomprehensible part of the article is the sample preparation for biological tests. As I understood from the article POVs-1 and 2 are the insoluble coordination polymers. So, how did the authors do the MTT-test and so on for solid compounds? Is it being a dispersion? In which solvent? What is the particle size and PDI? How did the authors prove that these dispersions are stable and do not hydrolyze?

Also, on page 4, lines 114-117, the authors write “A prevalent viewpoint is that organic carboxylic acids, have the potential in antiproliferative field….”. How do carboxylic acids relate to this work?

Finally, the conclusion of high anticancer activity is inadequate. The authors only investigated some biological effects on HepG-2 cancer cells without similar experiments on normal cell lines. It is well known that POMs in general have high cytotoxicity and acute toxicity, and their high anticancer activity, as well as their applicability as a contrast agent for CT, cannot be declared without serious studies, which are not presented in this paper.

In summary, before publication the authors should change the title of the manuscript, rewrite the abstract and conclusion, add a discussion of the chemistry and crystal structure (with comparison to literature data), and add information on sample preparation for biological studies.

Some of the sentences are incorrect. For example page 2, lines 57-58

"In this work, we investigated two novel hybrid POVs framework structures, [Ni(1-vIM)4]2[V4O12]·H2O (POVs-1), [Mn(1-vIM)4]2[V4O12]·H2O (POVs-2), were synthesized via selecting 1-vIM to assemble with ammonium metavanadate and different metal ions under hydrothermal conditions."

Author Response

August 16th, 2023

Dear Ms. Joyce Liu:

Thank you for all your considerations in evaluating our manuscript “Two Imidazole-decorated Tetranuclear Polyoxovanadates Clusters: synthesis, crystal structures and human hepatocellular carcinomas (HepG-2) inhibitory activities" (molecules-2548113) and we thank the reviewers for their devoted time and efforts. We are happy that they find the content of quality and novelty to ensure its suitability as a full paper in Molecules. Please find enclosed a revision of the manuscript, referenced above. We appreciate the reviewers’ constructive comments and suggestions to improve the manuscript. In response, we have done our best to resolve all the issues as well as revising the language to enhance the quality of this work.

In the revised version, we have addressed the points that the referees made. Point-by-point responses in the order the comments appeared in your letter and in the referees’ comments are provided below with the corresponding edits that we made to the manuscript and supporting information. These edits are also highlighted in yellow in both the manuscript and supporting information to reflect the changes.

Moreover, the manuscript has undergone editing for proper English language, grammar, punctuation, spelling and overall style by a highly qualified English speaking editor at EditRun.

We wanted to thank you and the referees for the insightful comments and for taking the time to provide critical input. We believe this manuscript is improved by this process, and hope you agree.

Best regards,

Yours Sincerely,

Zhen Li

lizhenlcu@163.com

School of Chemistry and Chemical Engineering,

Liaocheng University

The list of changes or comments is shown as following:

#Reviewer

Comments:

The article entitled “Two Imidazole-decorated Tetranuclear Polyoxovanadates Clusters: synthesis, crystal structures and human hepatocellular carcinomas (HepG-2) inhibitory activities” by Fumei Shi with co-authors is devoted to the preparation of two new coordination polymers based on tetranuclear polyoxovanadate (POV) [V4O12]4–, Ni(II) or Mn(II) cations and 1-vinylimidazole (1-vIM) ligands. The authors characterized the compounds obtained using single-crystal structural analysis, infrared spectroscopy and elemental analysis. In addition, some biological properties of the coordination polymers, named as POVs-1 and POVs-2, were evaluated. In general, the article is interesting and can be published in Molecules, but it has a number of serious flaws that need to be corrected before publication.

Response: Thanks for your positive comment.

First of all, tetranuclear polyoxovanadates (referred to as Tetranuclear Polyoxovanadates Clusters in the title) and polyoxometalates (POMs) (page 1, line 40) are not cluster metal cluster complexes. According to F.A. Cotton (Inorg. Chem., 1964, 3, 1217), metal cluster complexes have covalent metal-metal bonds. POMs are polyatomic ions consisting of three or more transition metal oxyanions linked together by shared oxygen atoms. The term cluster is used incorrectly in the article. Also, POVs-1 and 2 are not imidazole-decorated POMs since imidazole-decorated POM is when imidazole coordinates to POM with formation of V-N bonds. Therefore, the title and some parts of the introduction need to be changed. 

Response: Thank you for insightful suggestion. We have corrected the term cluster in the article and changed ‘metal clusters’ to ‘metal oxygen cluster’ (please see: page 1, 1ine 40). Moreover, for imidazole-decorated POMs in the article, we have already changed the title, abstract and some parts of the introduction. We have rewritten the abstract section and the introduction to correct some inappropriate words. “Polyoxometalates (POMs) have unique structural characteristics and excellent physical and chemical properties, which have attracted great attention of scholars in anticancer and chemotherapy field. Herein, two novel polyoxovanadates (POVs) [M(1-vIM)4]2[V4O12]·H2O (M = Ni and Mn, denoted as POVs-1 and POVs-2, respectively, 1-vIM= 1-vinylimidazole), were successively synthesized and structurally characterized. The two POVs are isomeric, and their basic structural units are composed of one [V4O12]4- cluster, two [M(1-vIM)4]2+ and one water molecule.” (please see: the abstract, 1ines 10-15, page 1). And we have changed the title “Ni/Mn-complex-tethered tetranuclear polyoxovanadates: crystal structure and inhibitory activity on human hepatocellular car-cinoma (HepG-2)”.

  1. Second, the main purpose of the article, according to the authors, is to study the biological effects of the obtained compounds, but this looks strange. In the abstract the authors write that “Polyoxovanadates (POVs), as emerging biologically active inorganic metal oxides, have been selected as promising antitumor alternatives in suppression of tumor growth because of low toxicity towards the human body.” Where are the references for such a loud statement? I've never heard that POMs are non-toxic to the human body. Moreover, according to Introduction parts and the results obtained, POVs is quite biologically active and cytotoxic. So, it is not correct to announce that POVs have “low toxicity towards the human body”.

Response: Thank you for insightful suggestion. According to the previous results, the rich structural types of polyoxometalates make them have great anticancer potential and the possibility of becoming highly effective chemotherapeutic drugs (Coord. Chem. Rev., 2021, 477, 214143). According to your opinion, we have corrected the low toxicity of polyoxovanadates (POVs) to humans. The sentence "Polyoxovanadates (POVs), as emerging biologically active inorganic metal oxides, have been selected as promising antitumor alternatives in suppression of tumor growth because of low toxicity towards the human body." has been changed to "Polyoxometalates (POMs) have unique structural characteristics and excellent physical and chemical properties, which have attracted great attention of scholars in anticancer and chemotherapy field." 

  1. In addition, since the compounds POVs-1 and POVs-2 have been prepared for the first time, it is necessary to describe them in detail and to compare them with similar compounds at least form Ref. 19. There the authors also obtained compound similar compounds of Ni and Cu, namely [Ni2(1-vIM)7H2O][V4O12]·H2O and [Cu2(1-vIM)8][V4O12]·H2O. Moreover, the compound [Cu2(1-vIM)8][V4O12]·H2O has a similar formula to POVs-1 and 2 prepared here, but in a different crystal packing. The authors should compare and discuss the differences in synthesis and crystal structure between them.

Response: Thank you for insightful suggestion. According to your value advice, we have provided more detailed crystal synthesis process and crystal structure analysis section. Single-crystal X-ray structural analyses reveal that the cluster units of POVs-2 and POVs-1 are isostructural, differing only in the transition metal atom. The structural formula of POVs-2 is [Mn(1-vIM)4]2[V4O12]·H2O, crystallizing in the P21/n space group. POVs-2 comprises of two Mn2+ cations, one [V4O12]4- anion cluster, and eight 1-vIM lig-ands. The vanadium atoms within the [V4O12]4- anion cluster exihibit a distorted [VO4] tetrahedral geometry. All manganese atoms in POVs-2 adopts a six-coordinate octahe-dral geometry, which coordination involving four ligands via imidazole N atoms around the central Mn atom, a terminal O atom derived from the [VO4] tetrahedron, and a surface O atom. Bond lengths and angles of Mn–N/O, V–O, O/N–Mn–O/N and O–V–O are listed in Table S3. Similarly, an additional direction of interaction between ligand molecules and adjacent [V4O12]4- pseudo planes allows POVs-2 to form an intri-guing 3D supramolecular structure (Figure 2c). Within these 3D networks, four adja-cent [VO4] tetrahedra are linked through corner-sharing O atoms, creating [V4O12]4- clusters. Additionally, one terminal O atom from the [VO4] tetrahedron engages in weak interaction to connect with another imidazole ligand subunit, thus contributing to the establishment of a complete 3D supramolecular structure. Notably, compared to POVs-1, POVs-2 exhibite fair-sized cavities. Unlike the V-O-Mn coordination bond re-ported in literature for the 3D structure, the supramolecular structure of POV-2 relies solely on weak interactions, devoid of V-O-M coordination bonds. The square pores of this framework measure approximately 2.63 Å across, based on the dimensions of the largest pink sphere capable of fitting into them, as shown in Figure 2d. (Please see the section of structure analysis in the article).

  1. The most incomprehensible part of the article is the sample preparation for biological tests. As I understood from the article POVs-1and 2are the insoluble coordination polymers. So, how did the authors do the MTT-test and so on for solid compounds? Is it being a dispersion? In which solvent? What is the particle size and PDI? How did the authors prove that these dispersions are stable and do not hydrolyze?

Response: Thank you for your critical suggestions. And we have reworded experiments process in detail. Moreover, DMSO is very stable and the boiling point is above with 150 °C, which is difficult to decompose under general temperature conditions and does not volatilize. It is almost a universal solvent for cytology research. Please see the paper revised in red color as follow:

Both synthesized complexes, POVs-1 or POVs-2, were dissolved in DMSO to create a 10 mM stock solution for subsequent dilution into working concentrations. HepG-2 cell viability was assessed using the 3-(4,5-dimethylthiazol-2-yl)-2,5-diphenyltetrazolium bromide (MTT) assay. Cells were seeded in a 96-well plate at a density of 1×105 cells mL−1 and cultured with varying concentrations of the complex, a combination of NAC/complexes, or in the absence of treatment, in a dose- and time-dependent manner. The concentration ranged from 10 μM to 100 μM, and the incubation times were 24 h and 48 h. The culture medium used was DMEM medium (Hyclone), supplemented with 10% FCS, 2 mM L-glutamine, 100 U mL−1 penicillin and 100 µg mL−1 streptomycin. The cells were cultured in a humidified atmosphere with 5% CO2 at 37 °C, and a native group was included as a negative control.

To perform the MTT assay, a solution of MTT (5 mg mL−1) was added to each well and incubated for an additional 4 h at 37 °C with 5% CO2. After incubation, the supernatant was carefully discarded, and 100 μL DMSO was added to each well and mixed thoroughly to dissolve the formazan crystals. The absorbance of the plate was measured at 490 nm using a multi-well plate reader (Agilent synergy H1). The percentage of cell viability was calculated using the following formula: % Viability = [OD of treated cells−OD of blank/OD of control cells−OD of blank]×100. All experiments were independently conducted in triplicate.

  1. Also, on page 4, lines 114-117, the authors write “A prevalent viewpoint is that organic carboxylic acids, have the potential in antiproliferative field….”. How do carboxylic acids relate to this work?

Response: Thank you for insightful suggestion. We have corrected the mistake in the article. The sentences "A prevalent viewpoint is that organic carboxylic acids, have the potential in anti-proliferative field and being prospective anticancer agents, some attempts have been facilitated to scrutinize their capacity of proliferation inhibition of tumor cells and to uncover the principal molecular mechanisms of their anti-cancer activity [22]" we have been deleted.

  1. 6. Finally, the conclusion of high anticancer activity is inadequate. The authors only investigated some biological effects on HepG-2 cancer cells without similar experiments on normal cell lines. It is well known that POMs in general have high cytotoxicity and acute toxicity, and their high anticancer activity, as well as their applicability as a contrast agent for CT, cannot be declared without serious studies, which are not presented in this paper.

Response: Thank you for insightful suggestion. We totally agree that would be very valuable with similar experiments on normal cell lines for further clinical experiments. While currently we here aim to investigate the biological effects of the POVs on HepG-2 cells and it is enough to show the complexes’ bioactivity on cancer cells with POVs blank controller as well as NAC+ POVs group just as other papers. For example, the effect of the PMoV inhibiting MCF-7 cell line was compared between MCF-7 blank controller and MCF-7 in the presence of PMoV (J. Inorg. Biochem., 2015, 152, 74-81). Another example is that Cheng et. al. reported that the inhibitory effects against the proliferation of tumor cells were observed with untreated or TEA-V10 treated tumor cells. (Polyhedron., 2018, 155, 313-319).

As for the anticancer activity to other cancer and normal cell lines as well as their more possibility of application like a contrast agent for CT, which will be discussed in subsequent studies.

  1. 7. In summary, before publication the authors should change the title of the manuscript, rewrite the abstract and conclusion, add a discussion of the chemistry and crystal structure (with comparison to literature data), and add information on sample preparation for biological studies.

Response: Thank you for insightful suggestion. We have changed the title of the manuscript to "Ni/Mn-complex-tethered tetranuclear polyoxovanadates: crystal structure and inhibitory activity on human hepatocellular carcinoma (HepG-2)"; And we have rewritten and corrected the abstract and conclusion sections, added a discussion of the chemistry and crystal structure with comparison to literature data (Dalton Trans., 2020, 49: 14148-14157). And we have added information on sample preparation for biological studies (Please see the above revised 3.4. Determination of Cell Viability).

Both synthesized complexes, POVs-1 or POVs-2, were dissolved in DMSO to create a 10 mM stock solution for subsequent dilution into working concentrations. HepG-2 cell viability was assessed using the 3-(4,5-dimethylthiazol-2-yl)-2,5-diphenyltetrazolium bromide (MTT) assay. Cells were seeded in a 96-well plate at a density of 1×105 cells mL−1 and cultured with varying concentrations of the complex, a combination of NAC/complexes, or in the absence of treatment, in a dose- and time-dependent manner. The concentration ranged from 10 μM to 100 μM, and the incubation times were 24 h and 48 h. The culture medium used was DMEM medium (Hyclone), supplemented with 10% FCS, 2 mM L-glutamine, 100 U mL−1 penicillin and 100 µg mL−1 streptomycin. The cells were cultured in a humidified atmosphere with 5% CO2 at 37 °C, and a native group was included as a negative control.

To perform the MTT assay, a solution of MTT (5 mg mL−1) was added to each well and incubated for an additional 4 h at 37 °C with 5% CO2. After incubation, the supernatant was carefully discarded, and 100 μL DMSO was added to each well and mixed thoroughly to dissolve the formazan crystals. The absorbance of the plate was measured at 490 nm using a multi-well plate reader (Agilent synergy H1). The percentage of cell viability was calculated using the following formula: % Viability = [OD of treated cells−OD of blank/OD of control cells−OD of blank]×100. All experiments were independently conducted in triplicate.

  1. 8. Some of the sentences are incorrect. For example page 2, lines 57-58, "In this work, we investigated two novel hybrid POVs framework structures, [Ni(1-vIM)4]2[V4O12]·H2O (POVs-1), [Mn(1-vIM)4]2[V4O12]·H2O (POVs-2), were synthesized via selecting 1-vIM to assemble with ammonium metavanadate and different metal ions under hydrothermal conditions."

Response: Thank you for insightful suggestion. We have corrected the sentences to "In this work, we synthesized two novel inorganic-organic hybrid POVs structures, [M(1-vIM)4]2[V4O12]·H2O (designated as POVs-1 and POVs-2), by utilizing 1-vinylimidazole (1-vIM) as the N-donor ligand to assemble with ammonium metavanadate and various metal ions under standard conditions"

Reviewer 2 Report

Shi et al. report two new polyoxovanadates and assess their potential as anticancer agents. They have included a range of assays to determine the modes of action, and whilst the work is of interest, there are several corrections which are required. The compounds need to be screened on a wider range of cell lines and include a normal cell line, and the work needs to be placed in the context of other vanadium/ POV compounds. Overall, the English and grammar needs to be thoroughly checked, as there are many issues. Some wording, e.g., “dazzling blue” needs to be removed from the manuscript, and incorrect terms, e.g., “in vigorous mitochondria”, “response to the insult of Complex”, “verdict of MMP”, “emitting red–orange fluorescing” etc, need to be corrected and reworded. I have just picked out a few examples, but the issues occur in almost all paragraphs. Once these comments have been addressed, this manuscript could become acceptable for publication.

Introduction

-       POV needs to be defined and introduced before discussing the applications.

-       What is “neutral pH”? All of the activities need to be quantified in the text.

-       “However, due to the structural properties of POVs, the development of tetranuclear POVs framework structures for further anticancer activity has been rarely reported.” – this is not clear. Why have they not been reported “due to the structural properties”?

-       A figure of other POVs would be useful.

-       1-vIM should be redefined in the text.

Results

-       The discussion of the synthesis is missing for both POVs. These need to be discussed before XRD. Please include a reaction scheme and brief discussion of the protocol.

-       Then in the XRD section, the growth conditions are missing and the images in Figures 1 and 2 are not clear – please make this larger and easier to read.

-       The angles are discussed as being distorted but there is no cross-referencing to data tables for bond lengths and angles.

-       The authors did not provide the CCDC numbers or the cif checks – these need to be uploaded with the submission before further assessment of this work. 

-       The FT-IR need to be normalised. Why is there a drift in the spectra at higher wavenumbers?

-       The PXRD Figure numbers are incorrect in the main text.

-       The introductory sentences to the Pharmacology Evaluation section needs to be reworded. The sentence structure and grammar are not correct in most of this section.

-       “was about 60 μM and 40 μM for POVs-1 and POVs-2 respectively” – “about” is not accurate enough, and the IC50 values need to be given exact with the standard errors.

-       These values are poor for vanadium compounds, so what is the justification for the additional experiments and compound design? How do these values compare to other POMs/POVs?

-       The authors need to conduct the cell viability studies on other cell lines and at least one normal cell line. Having screening against just HepG-2 is not sufficient for publication.

-       Vanadium is much less toxic and can be incubated with the cells for longer periods, do these compounds induce concentration dependent cytotoxicity?

-       Figure 3 caption states, MDA cells – is this an error and should this be HepG-2?

-       How was apoptosis measured? (Figure 4b) Also, is this early and late-stage apoptosis combined? More details and discussion are required and this entire section is not clear.

-       The introduction of NAC in both the cell viability and staining experiments has not been explained – please add in a paragraph to explain (Figures 4 and 5).

-       “it gives rise to the release of different apoptogenic factors from mitochondria into the cytoplasm, and disturbed mitochondrial energy metabolism to evoke reactive oxygen species (ROS) burst via decoupling proton by free radical in the respiratory chain.” – where is the evidence of this?

-       Figures 5 and 6 are too blurry and the graphs are too small.

Materials

-       The synthesis of 1 and 2 needs to be reworded – also the consistency of using labels POV-1/POV-2 or 1/2 needs to be kept throughout.  

-       Why weren’t sodium pyruvate and L-glutamine added to the media?

-       The MTT assay is missing information on stock solvents, concentrations, and amount/concentration of MTT dye. What does “All experiment western blots were conducted in triplicate independent experiments” mean? Why are Western blots used?

-       How much DAPI was used per well? 

-       The MMP assay is missing all details and could not be repeated.

-       How much H2DCFDA dye per well? What size well plates?

-       All incubation times and the machines/microscopes and software information included. 

The English is of low standard throughout and needs to be significantly improved.

Author Response

August 16th, 2023

Dear Ms. Joyce Liu:

Thank you for all your considerations in evaluating our manuscript “Two Imidazole-decorated Tetranuclear Polyoxovanadates Clusters: synthesis, crystal structures and human hepatocellular carcinomas (HepG-2) inhibitory activities" (molecules-2548113) and we thank the reviewers for their devoted time and efforts. We are happy that they find the content of quality and novelty to ensure its suitability as a full paper in Molecules. Please find enclosed a revision of the manuscript, referenced above. We appreciate the reviewers’ constructive comments and suggestions to improve the manuscript. In response, we have done our best to resolve all the issues as well as revising the language to enhance the quality of this work.

In the revised version, we have addressed the points that the referees made. Point-by-point responses in the order the comments appeared in your letter and in the referees’ comments are provided below with the corresponding edits that we made to the manuscript and supporting information. These edits are also highlighted in yellow in both the manuscript and supporting information to reflect the changes.

Moreover, the manuscript has undergone editing for proper English language, grammar, punctuation, spelling and overall style by a highly qualified English speaking editor at EditRun.

We wanted to thank you and the referees for the insightful comments and for taking the time to provide critical input. We believe this manuscript is improved by this process, and hope you agree.

Best regards,

Yours Sincerely,

Zhen Li

lizhenlcu@163.com

School of Chemistry and Chemical Engineering,

Liaocheng University

The list of changes or comments is shown as following:

#Reviewer 1

Comments:

Shi et al. report two new polyoxovanadates and assess their potential as anticancer agents. They have included a range of assays to determine the modes of action, and whilst the work is of interest, there are several corrections which are required. The s need to be screened on a wider range of cell lines and include a normal cell line, and the work needs to be placed in the context of other vanadium/ POV compounds. Overall, the English and grammar needs to be thoroughly checked, as there are many issues. Some wording, e.g., “dazzling blue” needs to be removed from the manuscript, and incorrect terms, e.g., “in vigorous mitochondria”, “response to the insult of Complex”, “verdict of MMP”, “emitting red–orange fluorescing” etc, need to be corrected and reworded. I have just picked out a few examples, but the issues occur in almost all paragraphs. Once these comments have been addressed, this manuscript could become acceptable for publication.

Response: Thanks for your insightful suggestion, we have added the activity comparisons with TBA-V10 compounds based on literature data (Polyhedron, 2018, 155: 313-319) suggesting that POVs-2 has better anticancer activity (please see the article, page 4, line 133-134).

And also we apologize for the unclear expressions in the last version. We have corrected and polished all the terms as below, please see details highlighted in yellow of the "Results and Discussion".

As shown in Figure 5a, the mitochondrial membrane potential (ΔΨm) was assessed using the specific fluorescence switch of the JC-1 probe. In healthy cells, JC-1 accumulates in the mitochondria cells as aggregates emitting red fluoresce. However, in treated cells, JC-1 remains in the cytoplasm in its monomeric form, emitting green fluoresce, indicating the collapse of mitochondrial membrane potential (ΔΨm). This transition from red to green fluorescence reflects JC-1's shift from an aggregator at normal ΔΨm to a monomer at collapsed ΔΨm. Depolarized or dysfunctional mitochondria, with reduced membrane potential, fail to uptake JC-1 and form aggregates. Similarly, the disturbed red and green fluorescence of mitochondria induced by complex POVs were significantly alleviated in the presence of NAC, which indicated that both complexes, POVs-1 and POVs-2, triggered this loss of ΔΨm, a crucial aspect of this investigation (Figure 5b). It gives rise to the release of different apoptogenic factors from mitochondria.

  1. POV needs to be defined and introduced before discussing the applications.

Response: Thank you for insightful suggestion. We have modified the definition of POMs in the introduction. Polyoxometalates (POMs) are a class of nano-sized metal-oxide polyoxoanion clusters, composed of corner- and edge-sharing [MOx] polyhedral, where M generally represents Mo, W, V, Nb, or Ta. POMs, as an adaptable nanoscale multinuclear metal-oxygen cluster, have captured tremendous attention across diverse fields, including biology, catalysis, energy, materials, and magnetism due to their well-defined structure, and adjustable physicochemical properties. (please see the article, lines 41-47, page 1)

  1. What is “neutral pH”? All of the activities need to be quantified in the text.

Response: Thank you for insightful suggestion. We have changed the “neutral pH” to “pH = 7.02”.

  1. “However, due to the structural properties of POVs, the development of tetranuclear POVs framework structures for further anticancer activity has been rarely reported.” – this is not clear. Why have they not been reported “due to the structural properties”?

Response: Thank you for insightful question. To date, vanadium substituted POMs and decavanadate-based complexes exhibits some degree of anticancer activities. Moreover,vanadium shows common oxidation states between +2 and +5, and exhibits a particularly rich coordination chemistry in aqueous solutions. POVs usually exhibit cage-, sphere-, hollow-, basket-, belt-, and barrel-like structural motifs, which are constructed of a number of polyhedra fused and/or linked through a common vertex (corner) and/or polyhedral edges and faces. These polyhedra consist of the homo- or heterovalent V atoms showing e.g. square-pyramidal [VO5], octahedral [VO6] and tetrahedral [VO4] coordination geometries. The tetranuclear [V4O12] clusters can easily assemble with organic ligands to form ordered framework structure, which may be beneficial to improve the anticancer activity of POVs.

  1. A figure of other POVs would be useful.

Response: Thank you for insightful suggestion. According to reports in the literature, we have compared the biological activity of POVs-1, POVs-2 and TBA-V10 compounds (the IC50 value = 76.7 ± 7.6 μg/mL) (Polyhedron, 2018, 155: 313-319) against HepG-2, and the comparison showed that POVs-2 (the IC50 value = 40 ± 6.51 μM) was more effective.

  1. 1-vIM should be redefined in the text.

Response: Thank you for insightful suggestion. We have redefined the 1-vIM in the abstract. (please see, line 13, page 1)

  1. The discussion of the synthesis is missing for both POVs. These need to be discussed before XRD. Please include a reaction scheme and brief discussion of the protocol.

Response: Thank you for insightful suggestion. We have added the scheme (Scheme 1) and the brief discussion in the article. (Please see the article, lines 70-73, and 76-78, page 2). The specific synthesis steps are described in the synthesis section.

Scheme 1. The controlling synthesis of POVs 12. All hydrogen atoms are omitted for clarity.

  1. Then in the XRD section, the growth conditions are missing and the images in Figures 1 and 2 are not clear – please make this larger and easier to read.

Response: Thank you for insightful suggestion. We have added the growth conditions about POVs-1 and POVs-2. The block crystals of POVs were obtained by using the conventional growth methods, which the reaction solution was slowly evaporated at room temperature for two weeks. The specific synthesis steps are described in the synthesis section. And we have corrected the images in Figure 1 and 2, ensuring that the images size are larger and clearer.

Preparation of POVs-1: VO(acac)2 (0.27 mmol, 0.072 g) and NiCl2·6H2O (0.25 mmol, 0.032 g) were added to 10mL of water. Then the tetramethylammonium hydroxide 25% water solution (0.1g) and 1-vinylimidazole (2 mL) were stirred for 1 h. Then the mixed solution was heated and stirred at 90 °C for 72 h. The above solution was thermally filtered, and the resulting filtrate slow evaporated at room temperature for one week, the pale blue block crystals of POVs-1 were obtained. The crystals suitable for SXRD analysis grew after leaving the solution to stand for two weeks.

Preparation of POVs-2: The synthetic procedure was similar to that of POVs-1, except that] NiCl2·6H2O (0.032 g, 0.25 mmol) was used to replace MnCl2·4H2O (0.031 g, 0.25 mmol). The brown crystals suitable for SXRD analysis grew after leaving the solution to stand for two weeks.

Figure 1. (a) Ball-and-stick packing structure representation of POVs-1; (b) The 2D network in POVs-1; (c) The 3D supramolecular structure of POVs-1; (d) the vIM ligands have been omitted in order to highlight the three-dimensional metal oxide substructure. Colour code: V, green; Ni, turquiose; O, red; N, blue; C, gray; H, white.

Figure 2. (a) Ball-and-stick packing structure representation of POVs-2; (b) The 2D network in POVs-2; (c) The 3D supramolecular structure of POVs-2; (d) the vIM ligands have been omitted in order to highlight the three-dimensional metal oxide substructure. Color code: V, green; Mn, yellow; O, red; N, blue; C, gray; H, white.

  1. The angles are discussed as being distorted but there is no cross-referencing to data tables for bond lengths and angles.

Response: Thank you for insightful suggestion. We have added cross-referencing to data tables for bond lengths and angles. We have added the sentences "Two imidazole complexes are arranged alternately and linked by [V4O12] cluster to form the polyoxovadanate POVs-1 with the V···V distance of 5.041 Å. The bond lengths and angles of Ni–N/O, V–O, O/N–Ni–O/N and O–V–O are listed in Table S2. Bond lengths and angles of Mn–N/O, V–O, O/N–Mn–O/N and O–V–O are listed in Table S3." to the structure analysis section.

  1. The authors did not provide the CCDC numbers or the cif checks – these need to be uploaded with the submission before further assessment of this work.

Response: Thank you for insightful suggestion. We have provided the CCDC numbers in Table S1, the checkcif about POVs-1 and 2 were uploaded.

  1. The FT-IR need to be normalised. Why is there a drift in the spectra at higher wavenumbers?

Response: Thank you for insightful suggestion. We have corrected the FT-IR spectra in Figure S1 and S2. There are many stretching vibration peaks with relatively small intensity in the spectra at higher wavenumbers. The small stretch bands about 2800 cm−1 are assigned to C-H (1-vIM ligand).

Figure. S1 The FT-IR spectrum of POVs-1.

Figure. S2 The FT-IR spectrum of POVs-2.

  1. The PXRD Figure numbers are incorrect in the main text.

Response: Thank you for insightful suggestion. We have corrected the PXRD Figure numbers.

  1. The introductory sentences to the Pharmacology Evaluation section needs to be reworded. The sentence structure and grammar are not correct in most of this section.

Response: Thank you very much for your suggestion. We have rewritten the sentence and corrected the grammatical errors in the article. The sentences "A prevalent viewpoint is that organic carboxylic acids, have the potential in anti-proliferative field and being prospective anticancer agents, some attempts have been facilitated to scrutinize their capacity of proliferation inhibition of tumor cells and to uncover the principal molecular mechanisms of their anti-cancer activity [22]" have been deleted.

  1. These values are poor for vanadium compounds, so what is the justification for the additional experiments and compound design? How do these values compare to other POMs/POVs?

Response: Thank you for insightful suggestion. According to reports in the literature, we have compared the biological activities of POVs-1, POVs-2 and TBA-V10 compounds (the IC50 value = 76.7 ± 7.6 μg/mL) (Polyhedron, 2018, 155: 313-319) against HePG2, and the comparison showed that POVs-2 (the IC50 value = 40 ± 6.51 μM) was more effective. And we will further design and synthesize new vanadium complexes, and compare their anticancer activity in more cell lines with that of previously reported POMs.

  1. 13.The authors need to conduct the cell viability studies on other cell lines and at least one normal cell line. Having screening against just HepG-2 is not sufficient for publication.

Response: Thank you for insightful suggestion. We totally agree your opinion. And more jobs such as concentration dependent cytotoxicity are undergoing and will be discussed in subsequent studies. We aim to investigate the biological activities of the complexes POVs on HepG-2 cells and it is enough to show bioactivity of the complexes on cancer cells with POVs blank controller just as other papers. Here are some examples, the effect of the PMoV inhibiting MCF-7 cell line was compared between MCF-7 blank controller and MCF-7 in the presence of PMoV (J. Inorg. Biochem., 2015, 152, 74-81). Another example is that Min Cheng et. al. reported that the inhibitory effects against the proliferation of tumor cells were observed with untreated or TEA-V10 treated tumor cells (Polyhedron., 2018, 155, 313-319).

  1. Vanadium is much less toxic and can be incubated with the cells for longer periods, do these compounds induce concentration dependent cytotoxicity?

Response: In order to explore whether the complex is concentration-dependent, the appropriate concentration gradient would generally be selected to explore the change of cytoinhibition rate. When a certain concentration is reached (within a certain time), The cytoinhibition rate no longer changes and prolonged action time is needed to explore whether the concentration dependence. In our investigation, see "Determination of Cell Viability" as below:

HepG-2 cell viability was assessed using the 3-(4,5-dimethylthiazol-2-yl)-2,5-diphenyltetrazolium bromide (MTT) assay. Cells were seeded in a 96-well plate at a density of 1×105 cells mL−1 and cultured with varying concentrations of the complex, a combination of NAC/complexes, or in the absence of treatment, in a dose- and time-dependent manner. The concentration ranged from 10 μM to 100 μM, and the incubation times were 24 h and 48h. The culture medium used was DMEM medium (Hyclone), supplemented with 10% FCS, 2 mM L-glutamine, 100 U mL−1 penicillin and 100 µg mL−1 streptomycin. The cells were cultured in a humidified atmosphere with 5% CO2 at 37 °C, and a native group was included as a negative control.

  1. Figure 3 caption states, MDA cells – is this an error and should this be HepG-2?

Response: Yes, we are SORRY for the mistake and have corrected into HepG-2 cells.

  1. 1 How was apoptosis measured? (Figure 4b) Also, is this early and late-stage apoptosis combined? More details and discussion are required and this entire section is not clear.

Response: As the results showed that both the two new hybrid complexes POVs-1 and POVs-2 could induce HepG-2 cells apoptosis. In order to more intuitively and clearly present the percentage of apoptosis cells induced by complexes, apoptosis cells were counted and analyzed via image J and supporting software in three parallel random sampling of the experimental results graph. Significant differences among treatment effects were determined using one-way ANOVA, followed by Tukey’s post hoc test for multiple comparisons, with statistical significance of p < 0.05. Please see the article, lines 360-365, page 11.

  1. 1 The introduction of NAC in both the cell viability and staining experiments has not been explained – please add in a paragraph to explain (Figures 4 and 5).

Response: Thank you very much for your suggestion. We have explained as follow:Furthermore, the intensified bright blue fluorescence and the percentage of apoptotic cells in the POVs-2 treated groups were notably diminished in the presence of NAC(N-Acetylcysteine), a classic antioxidant. This implies that both complexes, POVs-1 and POVs-2, induce apoptosis in HepG-2 cells by triggering intrinsic reactive oxygen species (ROS), with POVs-2 demonstrating stronger effects than POVs-1. Please see details in the third paragraph of the "2.2. Pharmacology Evaluation" (Figures 4).

Similarly, the disturbed red and green fluorescence of mitochondria induced by complex POVs were significantly alleviated in the presence of NAC, which indicated that both complexes, POVs-1 and POVs-2, triggered this loss of ΔΨm, a crucial aspect of this investigation (Figure 5b). Please see details in the fourth paragraph of the "2.2. Pharmacology Evaluation" (Figures 5).

  1. 1 “it gives rise to the release of different apoptogenic factors from mitochondria into the cytoplasm, and disturbed mitochondrial energy metabolism to evoke reactive oxygen species (ROS) burst via decoupling proton by free radical in the respiratory chain.” – where is the evidence of this?

Response: Thank you for insightful suggestion. About the mechanism of mitochondria related apoptosis mediated by ROS were proved by specific dye. JC-1 dye could indicate the interference of mitochondrial energy metabolism, and apoptosis causes ROS, and the relative amount of ROS caused by apoptogenic factors can be demonstrated by DCFH-DA probe. We have added the relevant literatures to the main text. Please see the references [29-33] in the article. The supported evidences were listed follow:

  1. Suzuki, Y.; Imai, Y.; Nakayama, H.; Takahashi, K.; Takio, K.; Takahashi, R. A. Serine Protease, HtrA2, Is Released from the Mitochondria and Interacts with XIAP, Inducing Cell Death. Cell, 2001, 8, 613–621.
  2. Arnoult, D.; Gaume, B.; Karbowski, M.; Sharpe, J.C.; Cecconi, F.; Youle, R.J. Mitochondrial release of AIF and EndoG requires caspase activation downstream of BAX/BAK-mediated permeabilization. EMBO J., 2003, 22,4385–4399.
  3. Otera, H.; Ohsakaya, S.; Nagaura, Z.; Ishihara, N.; Mihara, K. Export of mitochondrial AIF in response to proapoptotic stimuli depends on processing at the intermembrane space. EMBO J., 2005, 24. 1375–1386.
  4. Uren, R.T.; Dewson, G.; Bonzon, C.; Lithgow, T.; Newmeyer, D.D.; Kluck, R.M. Mitochondrial release of pro-apoptotic proteins: electrostatic interactions can hold cytochrome cbut not Smac/DIABLO to mitochondrial membranes. Biol. Chem., 2005, 280, 2266–2274.
  5. David, L.V. Apoptogenic factors released from mitochondria. BBA-Mol. Cell Res.2011, 1813, 546-550.

  1. 1 Figures 5 and 6 are too blurry and the graphs are too small.

Response: Thank you again. According to the request, we have replaced larger and clearer figures, which are enough for the identification of probe fluorescence.

  1. The synthesis of 1 and 2 needs to be reworded – also the consistency of using labels POV-1/POV-2 or 1/2 needs to be kept throughout.

Response: Thank you for insightful suggestion. We have reworded the synthesis section of POVs-1 and POVs-2 (please see the 3.2 Synthesis of POVs-1 and POVs -2). And we have corrected the consistency of using labels, we have kept the labels of POVs-1 and POVs-2. Please see the 3.2. Synthesis of POVs-1 and POVs -2.

Preparation of POVs-1: VO(acac)2 (0.27 mmol, 0.07 g) and NiCl2·6H2O (0.25 mmol, 0.032 g) were added to 10mL of water. The mixture was then stirred along with a 25% aqueous solution of tetramethylammonium hydroxide (0.1g) and 1-vinylimidazole (2 mL) for 1 h. Subsequently, the mixed solution was heated and stirred at 90 °C for 72 h. The resulting solution was then subjected to thermal filtration, and the obtained filtrate was allowed to slowly evaporate at room temperature over the course of one week, resulting in the formation of pale blue block crystals of POVs-1. Crystals suitable for SXRD analysis were obtained after leaving the solution undisturbed for two weeks. The overall yield was 53% (based on V). The calculated (found) elemental composition for C40H50Ni2N16O15V4 was as follows: C, 36.50 (36.39); H, 3.80 (3.93); N, 17.03 (16.87).

Preparation of POVs-2: The synthetic procedure for POVs-2 was similar to that of POVs-1 with the exception that NiCl2·6H2O (0.032 g, 0.25 mmol) was used instead of MnCl2·4H2O (0.031 g, 0.25 mmol). The resulting brown crystals suitable for SXRD analysis were obtained after allowing the solution to stand undisturbed for two weeks. The overall yield was 55% (based on V). The calculated (found) elemental composition for C40H50Mn2N16O15V4 was as follows: C, 36.71 (36.58); H, 3.82 (3.97); N, 17.13 (17.01).

  1. Why weren’t sodium pyruvate and L-glutamine added to the media? The MTT assay is missing information on stock solvents, concentrations, and amount/concentration of MTT dye. What does “All experiment western blots were conducted in triplicate independent experiments” mean? Why are Western blots used?How much DAPI was used per well? The MMP assay is missing all details and could not be repeated.How much H2DCFDA dye per well? What size well plates? All incubation times and the machines/microscopes and software information included.

Response: Thank you for insightful suggestion. All required experimental details have been completed, as detailed in modifications below.

3.4. Determination of Cell Viability

Both synthesized complexes, POVs-1 or POVs-2, were dissolved in DMSO to create a 10 mM stock solution for subsequent dilution into working concentrations. HepG-2 cell viability was assessed using the 3-(4,5-dimethylthiazol-2-yl)-2,5-diphenyltetrazolium bromide (MTT) assay. Cells were seeded in a 96-well plate at a density of 1×105 cells mL−1 and cultured with varying concentrations of the complex, a combination of NAC/complexes, or in the absence of treatment, in a dose- and time-dependent manner. The concentration ranged from 10 μM to 100 μM, and the incubation times were 24 h and 48 h. The culture medium used was DMEM medium (Hyclone), supplemented with 10% FCS, 2 mM L-glutamine, 100 U mL−1 penicillin and 100 µg mL−1 streptomycin. The cells were cultured in a humidified atmosphere with 5% CO2 at 37 °C, and a native group was included as a negative control.

To perform the MTT assay, a solution of MTT (5 mg mL−1) was added to each well and incubated for an additional 4 h at 37 °C with 5% CO2. After incubation, the supernatant was carefully discarded, and 100 μL DMSO was added to each well and mixed thoroughly to dissolve the formazan crystals. The absorbance of the plate was measured at 490 nm using a multi-well plate reader (Agilent synergy H1). The percentage of cell viability was calculated using the following formula: % Viability = [OD of treated cells−OD of blank/OD of control cells−OD of blank]×100. All experiments were independently conducted in triplicate.

3.5. DAPI Staining of Cells

A total of 1 × 105 cells were cultured in a six-well plate and treated with either POVs-1 or POVs-2 for 24h. Subsequently, the cells were fixed with 4% methanol for 30 min at room temperature and then rinsed twice with PBS buffer. Following this, the cells were stained with a 100 μL solution of DAPI (5 µg/mL) for 10 min in the dark. Then cells were then washed with PBS and examined under a Nikon Ti inverted fluorescent microscope equipped with NIS-Elements software for further images analysis. Cells displaying condensed and fragmented nuclei, with enhanced fluorescence upon DAPI staining, were identified as apoptotic cells [44]. Experiments were repeated three times.  

3.6. Assessment of Mitochondrial Membrane Potential (Δψm)

A total of 1 × 105 cells were treated with either POVs-1 or POVs-2 for 24 h. Following the treatment, the cells were washed twice with PBS and then suspended in a mitochondrial incubation buffer. JC-1, a lipophilic molecular probe that specifically penetrates the cell and accumulates in mitochondria, was added to the cells at a final concentration of 10 nM. The cells were then incubated at 37°C in the dark for 30 min. Afterwards, the mitochondrial membrane potential was determined using an inverted fluorescence microscope (Nikon Ti Japan) at an excitation wavelength of 495 nm and emission maxima at 514–529 nm for the monomer form, and at 585–590 nm for the J-aggregate form, as per the protocol. The intensity of fluorescence was analyzed using image J, and further image analysis was performed using the coupled NIS-Elements software of Nikon Ti inverted fluorescent microscope. These experiments were repeated three times.

3.7. Measurement of Intracellular Reactive Oxygen Species (ROS)

To assess the ability of complexes POVs-1 or POVs-2 to trigger ROS in HepG-2 cells, the cell-permeable fluorescent probe H2DCFDA was utilized. In brief, a total of 2 × 106 cells were treated with complex POVs-1 or POVs-2, or a combination of NAC and the complex in a CO2 incubator for 24 h. Subsequently, the medium was replaced with fresh medium containing 5 μg/mL of H2DCFDA, and the cells were further incubated for 30 min. After incubation, the cells were rinsed twice with chilled PBS. The green fluorescence intensity within the cells was then examined using an inverted fluorescence microscope (Nikon Ti Japan) equipped with excitation and emission filters set at 492–495 and 517–527 nm, respectively, as per the protocol. The fluorescence intensity was quantified with image J. These experiments were repeated three times.

Reviewer 3 Report

The new compounds are elegant examples of complex POV materials.  However, the suggested application is very doubtful and described in a very careless manner.

Some sentences are messy and incomprehensible e.g.: 

"POVs-2 consists of one tetranuclear 16 [V4O12]4- clusters, two [Mn(1-vIM)4]2+ and one dissociative water molecules." 

POVs-2 consists of one...clusters and one ...water molecules?

What is "dissociative water molecule"? Solvating water?

What about POVs-1, why isn't it described in the abstract? Since the compounds have similar composition and structure (except for the metal ions), they should be described collectively, with the letter M replacing Mn/ Ni. By the way they should be denoted as POV-1 and POV-2 and the acronym POVs should be used when both polyoxovanadates are described (plural form).

The experimental part is unacceptably careless. The description of the synthesis, which is referred to as hydrothermal, does not mention water:

"Preparation of POVs-1: VO(acac)2 (2 mmol) and nickel chloride (1 mmol) in 1-vinylimidazole (10 mL) were allowed to be sealed in a 50 mL Teflon-lined stainless steel container, which was heated to 90 °C under autogenously pressure for 72 hours."

Water must have been there as it is needed for the hydrolysis of VO(acac)2, formation of polyoxovanadate and formation of the crystal structure (these are hydrates!)

How were the compounds added to the culture medium for the cytotoxicity experiments? It is not described at all. "After treatment with POVs-1 or compound POVs-2 at various concentrations for different times..." The compounds are 2D coordination polymers - do they dissolve in water? And more importantly are they still the same when they dissolve? I really doubt it especially taking into account complex, pH-dependent equilibria in the aqueous solutions of oxovanadates.

With the regard to the above Fig. 3 captions states "Cells were treated with indicated Complex for 24 h as described in method section." No, it was not described in method section at all - neither times nor concentrations or solvent used to prepare the solutions.

I do not agree ith the sentence: "The diffraction peak positions of the experimental PXRD patterns of POVs-1 and POVs-2 are in good agreement with the simulated patterns from single crystal analysis, which indicate that the phase purity of bulk powders is satisfactory." There are discrepancies between the calculated and experimental PXRD pattern for POV-2 indicating crystalline impurities.

Please, rewrite the conclusions. You have tested the compounds with the use of a single cell line and you know nothing about the selectivity. You cannot write that these compounds are "promising antitumor drug candidates"

The authors mix singular and plural form and use (there are examples in the previous part), confuse the meaning of words and build very complex sentences with little meaning such as: "A prevalent viewpoint is that organic carboxylic acids, have the potential in anti-proliferative field and being prospective anticancer agents, some attempts have been facilitated to scrutinize their capacity of proliferation inhibition of tumor cells and to uncover the principal molecular mechanisms of their anti-cancer activity [22]."

By the way, the mention of organic carboxylic acids at the beginning of 2.2. Pharmacology Evaluation is completely unclear to me. What do these acids have to do with the topic of their paper? What does it mean: "some attempts have been facilitated to scrutinize their capacity" - "some attempts have been undertaken"? 

It seems to me that the complexity of the sentences is meant to cover the lack of a reliable literature survey.

Author Response

August 16th, 2023

Dear Ms. Joyce Liu:

Thank you for all your considerations in evaluating our manuscript “Two Imidazole-decorated Tetranuclear Polyoxovanadates Clusters: synthesis, crystal structures and human hepatocellular carcinomas (HepG-2) inhibitory activities" (molecules-2548113) and we thank the reviewers for their devoted time and efforts. We are happy that they find the content of quality and novelty to ensure its suitability as a full paper in Molecules. Please find enclosed a revision of the manuscript, referenced above. We appreciate the reviewers’ constructive comments and suggestions to improve the manuscript. In response, we have done our best to resolve all the issues as well as revising the language to enhance the quality of this work.

In the revised version, we have addressed the points that the referees made. Point-by-point responses in the order the comments appeared in your letter and in the referees’ comments are provided below with the corresponding edits that we made to the manuscript and supporting information. These edits are also highlighted in yellow in both the manuscript and supporting information to reflect the changes.

Moreover, the manuscript has undergone editing for proper English language, grammar, punctuation, spelling and overall style by a highly qualified English speaking editor at EditRun.

We wanted to thank you and the referees for the insightful comments and for taking the time to provide critical input. We believe this manuscript is improved by this process, and hope you agree.

Best regards,

Yours Sincerely,

Zhen Li

lizhenlcu@163.com

School of Chemistry and Chemical Engineering,

Liaocheng University

The list of changes or comments is shown as following:

#Reviewer

Comments:

The new compounds are elegant examples of complex POV materials. However, the suggested application is very doubtful and described in a very careless manner.

Response: Thank you for insightful suggestion. We have corrected carefully in the article, and the application of POV in anticancer activity has been extensively reported in the literature.

  1. Some sentences are messy and incomprehensible e.g.: "POVs-2 consists of one tetranuclear 16 [V4O12]4-clusters, two [Mn(1-vIM)4]2+and one dissociative water molecules." POVs-2 consists of one...clusters and one ...water molecules? What is "dissociative water molecule"? Solvating water?

Response: Thanks for your insightful suggestion. We have corrected sentences to "The two POVs are isomeric, and their basic structural units are composed of one [V4O12]4- cluster, two [M(1-vIM)4]2+ and one water molecule. " The dissociative water molecule we want to express is the free water molecule. Here, free water refers to guest molecules that are not involved in the coordination of the crystal structure, that is, free water. Free water molecules are combined with the main body through hydrogen bonding to form the POVs structure.

  1. What about POVs-1, why isn't it described in the abstract? Since the compounds have similar composition and structure (except for the metal ions), they should be described collectively, with the letter M replacing Mn/ Ni. By the way they should be denoted as POV-1 and POV-2 and the acronym POVs should be used when both polyoxovanadates are described (plural form).

Response: Thanks for your insightful suggestion. We have rewritten the abstract, and we have described collectively complexes with the letter M replacing Mn/ Ni. " Herein, two novel polyoxovanadates (POVs) [M(1-vIM)4]2[V4O12]·H2O (M = Ni and Mn, denoted as POVs-1 and POVs-2, respectively, 1-vIM= 1-vinylimidazole), were successively synthesized and structurally characterized. The two POVs are isomeric, and their basic structural units are composed of one [V4O12]4- cluster, two [M(1-vIM)4]2+ and one water molecule. "

  1. The experimental part is unacceptably careless. The description of the synthesis, which is referred to as hydrothermal, does not mention water: "Preparation of POVs-1: VO(acac)2(2 mmol) and nickel chloride (1 mmol) in 1-vinylimidazole (10 mL) were allowed to be sealed in a 50 mL Teflon-lined stainless steel container, which was heated to 90 °C under autogenously pressure for 72 h." Water must have been there as it is needed for the hydrolysis of VO(acac)2, formation of polyoxovanadate and formation of the crystal structure (these are hydrates!)

Response: Thanks for your insightful suggestion. The polyoxovanadates, formation of the two crystal structures does require the presence of water. We added 10 mL water during the synthesis of our experiments. Therefore, we have rewritten the description of the synthesis, and the reaction solution is water. "Preparation of POVs-1: VO(acac)2 (0.27 mmol, 0.07 g) and NiCl2·6H2O (0.25 mmol, 0.032 g) were added to 10mL of water. The mixture was then stirred along with a 25% aqueous solution of tetramethylammonium hydroxide (0.1g) and 1-vinylimidazole (2 mL) for 1 h. Subsequently, the mixed solution was heated and stirred at 90 °C for 72 h. The resulting solution was then subjected to thermal filtration, and the obtained filtrate was allowed to slowly evaporate at room temperature over the course of one week, resulting in the formation of pale blue block crystals of POVs-1. Crystals suitable for SXRD analysis were obtained after leaving the solution undisturbed for two weeks." 

  1. How were the compounds added to the culture medium for the cytotoxicity experiments? It is not described at all. "After treatment with POVs-1 or compound POVs-2 at various concentrations for different times..." The compounds are 2D coordination polymers - do they dissolve in water? And more importantly are they still the same when they dissolve? I really doubt it especially taking into account complex, pH-dependent equilibria in the aqueous solutions of oxovanadates.

Response: Thanks for your insightful suggestion. We used the DMSO as solvent to dissolve the POVs, and added to the culture medium for the cytotoxicity experiments. They are the same when dissolved by DMSO. We don’t consider the pH-dependent equilibria in the aqueous solutions of oxovanadates. DMSO is almost a universal solvent for cytology research.

  1. With the regard to the above Fig. 3 captions states "Cells were treated with indicated Complex for 24 h as described in method section." No, it was not described in method section at all - neither times nor concentrations or solvent used to prepare the solutions.

Response: Thanks for your insightful suggestion. Both synthesized complexes, POVs-1 or POVs-2, were dissolved in DMSO to create a 10 mM stock solution for subsequent dilution into working concentrations. HepG-2 cell viability was assessed using the 3-(4,5-dimethylthiazol-2-yl)-2,5-diphenyltetrazolium bromide (MTT) assay. Cells were seeded in a 96-well plate at a density of 1×105 cells mL−1 and cultured with varying concentrations of the complex, a combination of NAC/complexes, or in the absence of treatment, in a dose- and time-dependent manner. The concentration ranged from 10 μM to 100 μM, and the incubation times were 24 h and 48 h. The culture medium used was DMEM medium (Hyclone), supplemented with 10% FCS, 2 mM L-glutamine, 100 U mL−1 penicillin and 100 µg mL−1 streptomycin. The cells were cultured in a humidified atmosphere with 5% CO2 at 37 °C, and a native group was included as a negative control.

To perform the MTT assay, a solution of MTT (5 mg mL−1) was added to each well and incubated for an additional 4 h at 37 °C with 5% CO2. After incubation, the supernatant was carefully discarded, and 100 μL DMSO was added to each well and mixed thoroughly to dissolve the formazan crystals. The absorbance of the plate was measured at 490 nm using a multi-well plate reader (Agilent synergy H1). The percentage of cell viability was calculated using the following formula: % Viability = [OD of treated cells−OD of blank/OD of control cells−OD of blank]×100. All experiments were independently conducted in triplicate.

  1. 5. I do not agree with the sentence: "The diffraction peak positions of the experimental PXRD patterns of POVs-1 and POVs-2 are in good agreement with the simulated patterns from single crystal analysis, which indicate that the phase purity of bulk powders is satisfactory." There are discrepancies between the calculated and experimental PXRD pattern for POV-2 indicating crystalline impurities.

Response: Thanks for your insightful suggestion. We have corrected the sentence to " The simulated and experimental PXRD patterns of POVs-1 and POVs-2 are shown in Figure S3 and Figure S4, respectively. The diffraction peak positions of the experimentally obtained PXRD patterns of POVs-1 and POVs-2 closely align with those of the simulated single crystal patterns." There are discrepancies indicating the presence of a small amounts of impurities, the analytical reason may be that we did not wash the crystals clean when we collected them. Please see the Result and Discussion 2.1.

  1. 6. Please, rewrite the conclusions. You have tested the compounds with the use of a single cell line and you know nothing about the selectivity. You cannot write that these compounds are "promising antitumor drug candidates"

Response: Thanks for your insightful suggestion. We have rewritten the conclusions and deleted the sentence about "promising antitumor drug candidates".

In this study, we have reported two intriguing POVs with distinct metal coordination modes, specifically the classical six-coordination mode that forms a two-dimensional grid. Notably, this metal coordination pattern has exhibited a significant inhibitory effect on HepG-2 cells. Both POVs-1 and POVs-2 have demonstrated robust antitumor effects against HepG-2 cells, achieved through the induction of mitochondrial-targeted intracellular ROS burst. This intriguing finding suggests their promising potential as candidates for the development of anti-cancer drugs. Moreover, the hybrid POVs-2 has shown even greater efficacy compared to POVs-1, which indicate a pivotal role played by the introduction of the Mn element, which confers superior biological activity compared to the Ni element. Therefore, additional experiments to accurately determine the IC50 value, thus assessing the degree of HepG-2 cells tolerance, are highly warranted. Furthermore, considering the potential clinical applications, it becomes imperative to conduct toxicological studies involving exposure to lower doses of POMs and their related complexes, especially in the context of in vivo CT imaging. This work provides a new perspective on the development of novel anticancer drugs by introducing bioactive Mn elements into the atomic level structure of POVs.

  1. 7. The authors mix singular and plural form and use (there are examples in the previous part), confuse the meaning of words and build very complex sentences with little meaning such as: "A prevalent viewpoint is that organic carboxylic acids, have the potential in anti-proliferative field and being prospective anticancer agents, some attempts have been facilitated to scrutinize their capacity of proliferation inhibition of tumor cells and to uncover the principal molecular mechanisms of their anti-cancer activity [22]." By the way, the mention of organic carboxylic acids at the beginning of 2. Pharmacology Evaluationis completely unclear to me. What do these acids have to do with the topic of their paper? What does it mean: "some attempts have been facilitated to scrutinize their capacity" - "some attempts have been undertaken"?It seems to me that the complexity of the sentences is meant to cover the lack of a reliable literature survey.

Response: Thanks for your insightful suggestion. We have corrected the grammatical errors in the article. The sentences "A prevalent viewpoint is that organic carboxylic acids, have the potential in anti-proliferative field and being prospective anticancer agents, some attempts have been facilitated to scrutinize their capacity of proliferation inhibition of tumor cells and to uncover the principal molecular mechanisms of their anti-cancer activity [22]" have been deleted.

Round 2

Reviewer 1 Report

The authors gave adequate comments and made requested corrections. In my opinion, now the article can be published in its current form.

Reviewer 3 Report

The manuscript can be accepted in its present form.